# Intraparietal stimulation disrupts negative distractor effects in human multi-alternative decision-making

Carmen Kohl[1,2], Michelle XM Wong[1], Jing Jun Wong[1], Matthew FS Rushworth[3], Bolton KH Chau[1,4]*

[1]Department of Rehabilitation Sciences, The Hong Kong Polytechnic University, Hong Kong, China; [2]Department Neuroscience, Carney Institute for Brain Sciences, Brown University, Providence, United States; [3]Department of Experimental Psychology, University of Oxford, Oxford, United Kingdom; [4]University Research Facility in Behavioral and Systems Neuroscience, The Hong Kong Polytechnic University, Hong Kong, China

**Abstract** There has been debate about whether addition of an irrelevant distractor option to an otherwise binary decision influences which of the two choices is taken. We show that disparate views on this question are reconciled if distractors exert two opposing but not mutually exclusive effects. Each effect predominates in a different part of decision space: (1) a positive distractor effect predicts high-value distractors improve decision-making; (2) a negative distractor effect, of the type associated with divisive normalisation models, entails decreased accuracy with increased distractor values. Here, we demonstrate both distractor effects coexist in human decision making but in different parts of a decision space defined by the choice values. We show disruption of the medial intraparietal area (MIP) by transcranial magnetic stimulation (TMS) increases positive distractor effects at the expense of negative distractor effects. Furthermore, individuals with larger MIP volumes are also less susceptible to the disruption induced by TMS. These findings also demonstrate a causal link between MIP and the impact of distractors on decision-making via divisive normalisation.

*For correspondence:
boltonchau@gmail.com

**Competing interest:** The authors declare that no competing interests exist.

## Editor's evaluation

This work presents fundamental findings elucidating the debate on how value-based choice behavior is influenced by seemingly irrelevant options (distractors). With convincing behavioral evidence following non-invasive brain stimulation, the authors provide support for the role of medial intraparietal cortex in divisive normalization in decision making. Given the importance of context effects as suboptimal violations of normative choice theories, this finding is significant and broadly relevant to psychologists, neuroscientists, and economists interested in decision making.

## Introduction

Making effective choices is a crucial part of human cognition. While binary choices have been studied extensively, there has been growing interest in complex decisions with more response alternatives (*Albantakis et al., 2012*; *Chung et al., 2017*; *Churchland et al., 2008*; *Kohl et al., 2019*; *Churchland*

*and Ditterich, 2012*). It has been claimed that introducing seemingly irrelevant 'distractor' alternatives influences decisions between the options of interest (*Chung et al., 2017*; *Chau et al., 2014*; *Spektor et al., 2018*). However, while most evidence suggests that relative preferences between two options crucially depend on the value of added distractor options (*Chau et al., 2014*; *Louie et al., 2013*; *Louie et al., 2011*), there is little consensus regarding the most fundamental feature of the nature of the distractor's influence – whether it makes decision-making worse or better – and the underlying neural mechanisms affecting these preferences.

One potential mechanism underlying this phenomenon is divisive normalisation. Based on the principles of normalisation observed in sensory systems (*Carandini and Heeger, 1994*; *Heeger, 1992*), Louie and colleagues (*Louie et al., 2011*) proposed that the encoded value of a given stimulus corresponds to the stimulus' absolute value divided by the weighted sum of the absolute values of co-occurring stimuli. Despite differences in how exactly divisive normalisation is formalized (e.g. noise and weight assumptions), in general the hypothesis suggests that neural responses during decision-making or accuracy in decision-making are increased as the value of the best option increases and as the values of the remaining alternatives decrease. This hypothesis receives empirical support in a number of studies in humans and monkeys (*Louie et al., 2013*; *Khaw et al., 2017*; *Louie et al., 2014*; *Pastor-Bernier and Cisek, 2011*; *Rorie et al., 2010*; *Webb et al., 2021*). Recently, there has been support for this view from experiments showing that divisive normalisation can be applicable to multi-attribute choices, as normalisation can occur at the level of individual attributes before they are combined into an overall option value (*Landry and Webb, 2021*; *Dumbalska et al., 2020*).

A seemingly opposing effect has been reported by *Chau et al., 2014*. They reported that human participants, who were asked to choose between two alternatives, displayed lower accuracy scores when a third low-value, rather than a third high-value option, was presented. Chau and colleagues modelled their behavioural findings using a biophysically plausible mutual inhibition model, in which different choice alternatives are represented by competing neural populations. Recurrent excitation within, and inhibition between populations create attractor dynamics, with one population displaying the highest firing rates, thereby indicating the choice of the associated response alternative (*Wang, 2002*). When a third option of high value, as opposed to low value, is added, however, inhibition between the populations is greater. This slows down the choice of the network such that it is less susceptible to noise and more capable of distinguishing between values, particularly when the value difference between the two available options is small. *Chau et al., 2014* reported just such a change in behaviour and further explored the mechanisms mediating the effect using functional magnetic resonance imaging (fMRI). Blood oxygen-level dependent signals associated with a key decision variable, the value difference between the two available options, in the ventromedial prefrontal cortex (vmPFC) were weaker when the distractor values were lower, further supporting the model predictions. Despite the fact that these findings appeared to stand in contrast to divisive normalisation, Chau and colleagues (*Chau et al., 2014*), also noted that neural activity in the medial intraparietal area (MIP) of the parietal cortex showed patterns consistent with divisive normalisation.

Recently, however, it has been argued that neither the negative distractor effects predicted by divisive normalisation models nor the positive distractor effects predicted by mutual inhibition models are robust (*Gluth et al., 2018*; *Gluth et al., 2020*). One way of reconciling these disparate points of view (positive distractor effects; negative distractor effects; no distractor effects), however, is the notion that, in fact, both positive and negative distractor effects occur but predominate to different degrees in different circumstances. This can be achieved by having a dual-route model that contains both a 'divisive normalisation' component and a 'mutual inhibition' component that run in parallel for making a decision in a race (*Chau et al., 2014*).

For example, careful consideration of the dual-route model suggests that the negative influence of the distractor should be most prominent in certain parts of a 'decision space' defined by two dimensions –the total sum and the difference in values of the options (*Chau et al., 2020*). Firstly, in the divisive normalisation component, the negative impact caused by variance in distractor value should be greatest when the total sum of option values is low. In accordance with this prediction, distractors reliably exert a significant and negative effect on decision-making, when the sum of the values of the choosable options is small (*Chau et al., 2020*). Secondly, positive distractor effects should predominate in the parts of the decision space in which the values of both choosable options are close and decisions are difficult but the opposite should happen when decisions are easy to make because the

choice option values are far apart. Both positive and negative distractor effects are apparent even in data sets in which they have been claimed to be absent (*Chau et al., 2014*; *Chau et al., 2020*).

Another prediction of this composite model is that if both types of distractor effect exist, then one might be promoted at the expense of the other by a manipulation that made decision-making relatively more dependent on different parts of the distributed neural circuit mediating decision-making. For example, if the negative distractor effects linked to divisive normalisation are associated with intraparietal cortical areas such as MIP, then disrupting MIP should decrease their prevalence at the expense of the positive distractor effects associated with other decision-making areas such as vmPFC.

In the current study, we, therefore, used transcranial magnetic stimulation (TMS) over MIP while human participants performed a value-based decision-making task. First, we predicted that, in the absence of any TMS, we would observe both significant positive and significant negative distractor effects but they would predominate in different parts of decision space. Second, based on previous findings (*Chau et al., 2014*; *Louie et al., 2013*), we hypothesised that disrupting MIP using TMS would decrease the effect of divisive normalisation in the decision-making process. This would reduce the negative distractor effect and increase the expression of the opposing positive distractor effect.

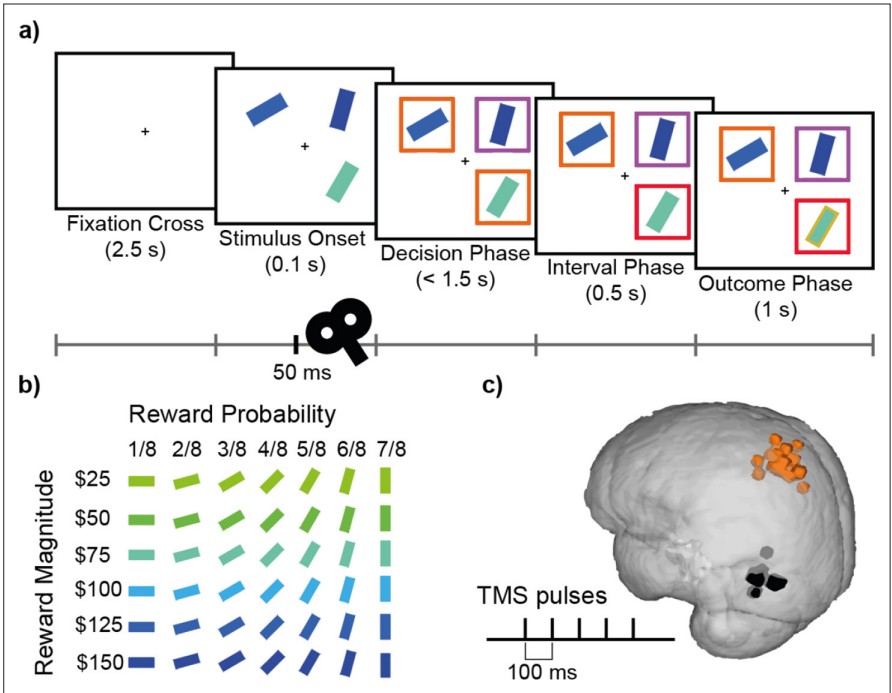

**Figure 1.** Decision-making task and TMS. (**a**) Participants completed a value-based decision-making task. Three rectangular stimuli representing three options were presented. After a brief period (0.1 s; *Stimulus Onset*), two of the stimuli were marked as choosable (labelled by orange boxes) while the third stimulus was marked as a distractor (labelled by a purple box). Participants had 1.5 s to indicate their decision (*Decision Phase*). The option that was chosen was highlighted with a red box (*Interval Phase*, 0.5 s). Subsequently, a gold/grey margin around the stimulus indicated whether or not they received the reward associated with their response (*Outcome Phase*; 1 s). (**b**) Each stimulus was defined by its colour and orientation, which indicated the associated reward magnitude and the probability of receiving the reward, respectively. Stimuli ranged in reward magnitude from $25 to $150, and in reward probability from 1/8 to 7/8. An example of the colour-magnitude and orientation-probability mappings is shown in **b**. The mappings were randomised across participants. (**c**) In each experimental session, repetitive TMS (5 pulses, 10 Hz) was applied over either the MIP or MT region in 1/3 of the trials. Orange and black highlights indicate MNI locations for MIP (average X=-35, Y=-53, Z=63) and MT (average X=-53, Y=-77, Z=5) stimulation sites for individual subjects, on a standard MNI brain.

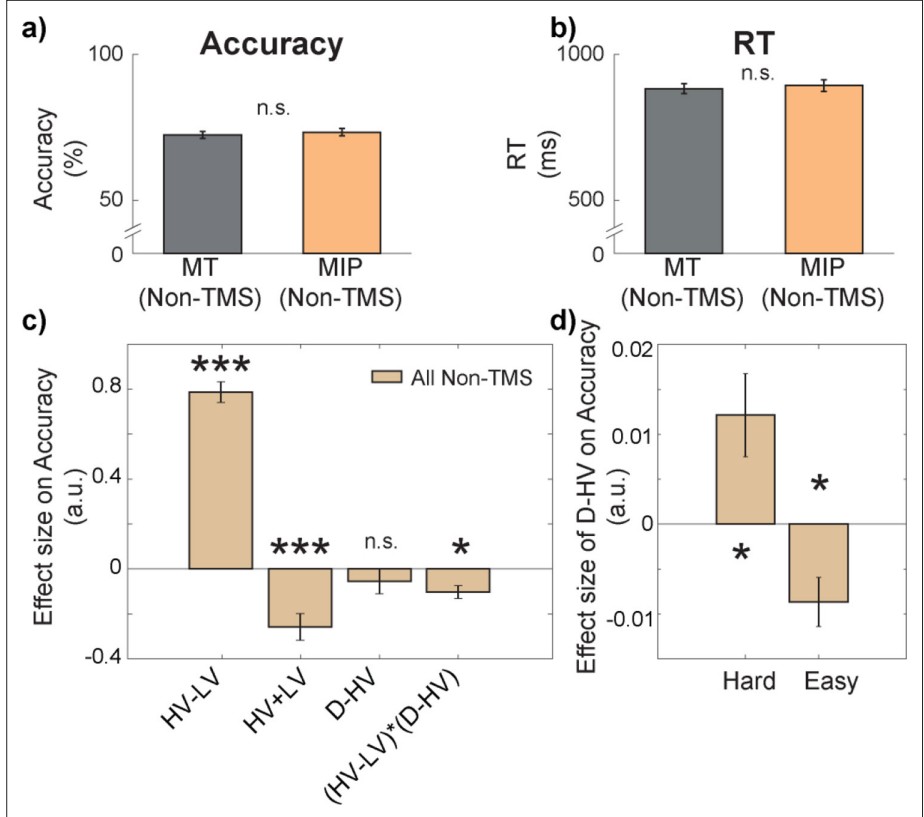

**Figure 2.** There was a negative distractor effect on accuracy in easy trials and a positive distractor effect on accuracy in hard trials. There were no differences in either (**a**) accuracy or (**b**) reaction time (RT) between Non-TMS trials in each session (MT/MIP). (**c**) GLM1 revealed that there was a negative (HV-LV)(**D–HV**) effect on accuracy, suggesting that the distractor effect (i.e. **D–HV**) varied as a function of difficulty (i.e. HV-LV). (**d**) A follow-up analysis on the (HV-LV)(**D–HV**) interaction using GLM2 showed that the distractor effect on accuracy was positive on hard trials and it was negative on easy trials. Error bars denote standard error. * p<0.050, *** p<0.001.

The online version of this article includes the following figure supplement(s) for figure 2:

**Figure supplement 1.** Participants showed smaller negative distractor effect when HV +LV was large.

**Figure supplement 2.** A third account suggests salient distractors can capture attention and eventually be chosen.

## Results

### Behavioural data: Increased positive distractor effect under contralateral MIP stimulation

A value-based decision-making task was used, in which participants were presented with two choosable options (high-value option HV or low-value option LV) and one distractor (D; *Figure 1*). As explained in the Methods, we looked at the impact of TMS to either MIP or a nearby control site (in the vicinity of area V5 or MT). However, both experiments also included an additional type of control condition: trials in which no TMS was applied.

First, we consider behavioural performance in the control situation in the absence of any TMS. To do this we combined data from Non-TMS trials of both MIP and MT sessions. All participants displayed above chance performance in both the main decision-making task, and the "matching" trials (additional trials that aimed to prevent participants from ignoring the identity of the distractor; see Methods), with an average accuracy of 72.84% (SD = 6.71; *Figure 2a*), and 70.14% (SD = 8.75) respectively, indicating that all participants followed task instructions. The average RT of the main task was 887.71ms (SD = 102.17ms; *Figure 2b*). There were no differences in either accuracy, $t(30)$ = 1.14, p=0.265, Cohen's d ($d$)=0.14, 95% confidence interval (CI) = [–0.01, 0.03], or reaction time (RT),

$t(30) = 0.64$, p=0.525, $d$=0.11, CI = [–23.61, 45.33], between Non-TMS trials in MIP and MT sessions (*Figure 2a and b*).

In general, larger distractor values should promote more accurate choices when decisions are hard (positive D-HV effect on trials with small HV-LV value difference) and, in contrast, they should impair choice accuracy when decisions are easy (negative D-HV effect on trials with large HV-LV value difference; *Chau et al., 2014*; *Chau et al., 2020*). In other words, there should be a negative (HV-LV)(D-HV) interaction effect. We tested whether this was the case in the current experiment by applying the same GLM (GLM1) as in *Chau et al., 2014*; *Chau et al., 2020*. In particular, it involved the following terms: the difference in value between the two available options (HV-LV), their sum (HV+LV), the difference between the distractor value and the high-value option (D-HV), and the interaction term (HV-LV)(D-HV). On Non-TMS trials, there was a positive HV-LV effect ($t(30) = 17.09$, p<0.001, $d$=3.07, CI = [0.69, 0.88]; *Figure 2c*) and a negative HV+LV effect ($t(30) = –4.35$, p<0.001, $d$=–0.78, CI = [-0.38,–0.14]), suggesting that more accurate choices were made on trials that were easier and consisted of options with poorer values. There was no D-HV effect ($t(30) = –0.95$, p=0.350, $d$=–0.17, CI = [–0.17, 0.06]) but critically there was a negative (HV-LV)(D-HV) interaction effect ($t(30) = –3.52$, p=0.001, $d$=–0.63, CI = [-0.16,–0.04]). To further examine the pattern of the negative (HV-LV)(D-HV) effect, we median split the data according to HV-LV levels and applied GLM2 to test the critical D-HV effect. On hard trials with small HV-LV, there was a positive D-HV effect ($t(30) = 2.62$, p=0.014, $d$=0.47, CI = [0, 0.02]; *Figure 2d*), whereas on easy trials with large HV-LV, there was a negative D-HV effect ($t(30) = –3.15$, p=0.004, $d$=–0.57, CI = [–0.01, 0]).

In addition, divisive normalisation models also predict that the size of the *negative* distractor effect should be *smaller* when the total HV+LV is *large* (*Chau et al., 2020*). This is because the variance in D then makes a smaller contribution to the overall normalisation effect that depends on HV+LV + D. If present, such an effect can be demonstrated by a positive (HV+LV)D interaction effect. This was indeed the case in the data of the current study (*Figure 2—figure supplement 1a*). One may argue that the distractor was only irrelevant to choices when its value is smallest. An additional analysis that excluded trials where the D exceeded the value of LV or HV confirmed that the distractor effect remained comparable (*Figure 2—figure supplement 1b*). In summary, these results are broadly consistent with recent demonstrations that both positive and negative distractor effects are reliable and statistically significant but that they predominate in different parts of the decision space (*Chau et al., 2014*; *Chau et al., 2020*).

Finally, we note that, in addition to the positive and negative distractor effects on choices between HV and LV, there is a third route by which the distractor can affect decision making – salient distractors can capture attention and eventually be chosen (*Gluth et al., 2018*; *Gluth et al., 2020*). In an additional analysis reported in *Figure 2—figure supplement 2*, we showed that an attentional capture effect was also present in our data.

Next, we examined whether MIP has any role in generating distractor effects by comparing TMS and Non-TMS trials. In addition, neurons in the intraparietal sulcus mostly have response fields in the contralateral side of space, for example, enhancing the posterior parietal cortex that includes the MIP using transcranial direct current stimulation can bias selection of choices that are presented on the contralateral side of space (*Woo et al., 2022*). Hence, we took care to consider whether any impact of TMS might be particularly robust when the distractor was presented contralateral to the MIP region that was targeted with TMS. We therefore split the trials according to whether the distractor was located on the contralateral side of space to the TMS. In other words, each analysis involved approximately one-fourth of the data that was split according to *Stimulation* (TMS/Non-TMS) and *Distractor Location* (ipsilateral/ contralateral side). Ideally, the trials should be split further according to difficulty, as indexed by the HV-LV difference, in order to isolate the negative distractor effect on easy trials that may be linked to MIP. However, that would mean that the analyses would rely on approximately one-eighth of the data and run the risk of becoming under-powered due to the small number of trials. Hence, we adapted GLM1 by removing the (HV-LV)(D-HV) term and keeping the remaining terms – HV-LV, HV+LV, D-HV (GLM3). We should now expect the absence of a D-HV main effect in the control Non-TMS data because the positive and negative distractor effects cancel out one another when they are no longer captured by a negative (HV-LV)(D-HV) interaction term. However, if TMS disrupts the negative distractor effect specifically and spares the positive distractor effect, then a positive D-HV effect should be revealed in the TMS data.

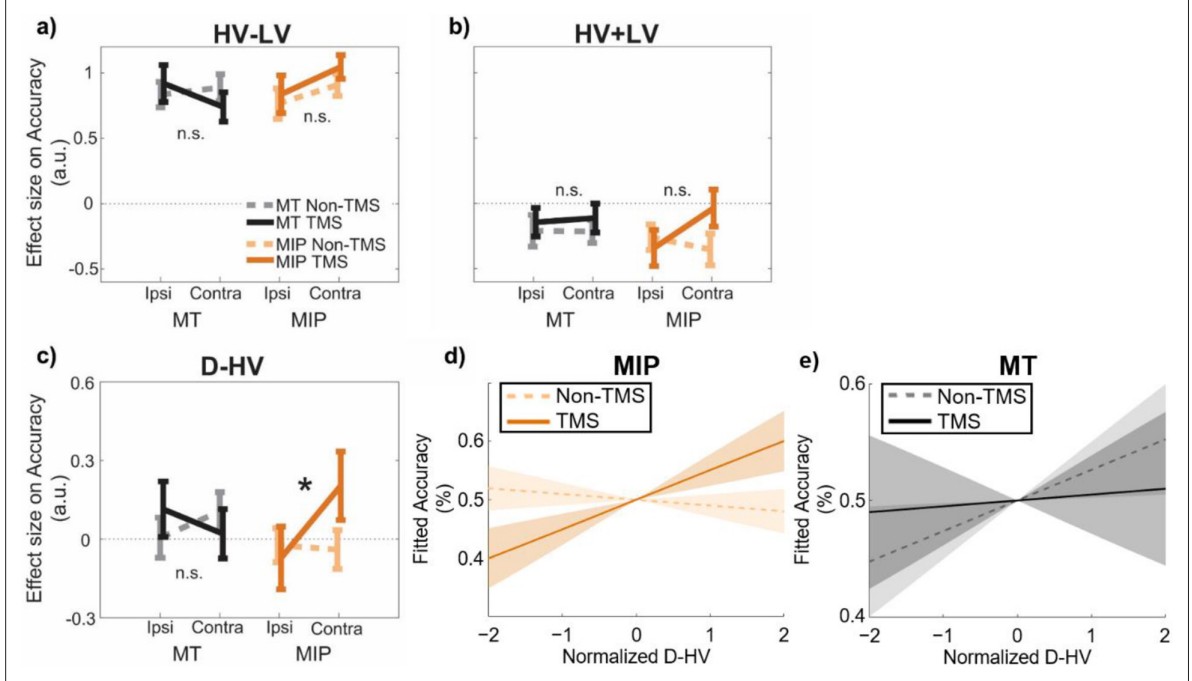

**Figure 3.** A positive distractor effect on choice accuracy was revealed after MIP was disrupted using TMS. GLM3 with the predictors (**a**) HV-LV, (**b**) HV +LV, and (**c**) D-HV was used to predict decision accuracy. Higher distractor values (i.e. higher D-HV) were associated with higher accuracy only when TMS was applied to MIP contralateral to the distractor, but not in any Non-TMS trials or in MT sessions: * p<0.050. It is important to note that '*' and 'n.s.' here denote significant and non-significant interaction (TMS x Site x D Location) effects respectively. The symbols are not intended to indicate whether other effects are or are not significant. For example, high HV-LV difference continues to predict accuracy as in *Figure 2c* (F(1,30) = 290.28, p<0.001, $\eta_p2$ = 0.91). In accordance with the divisive normalisation model, high HV +LV continues to predict low accuracy as *Figure 2c* (F(1,30) = 12.24, p=0.001, $\eta_p2$ = 0.29). (**d**) Fitted accuracy plotted as a function of D-HV when TMS (solid, orange) or no TMS (dotted, orange) was applied over the MPI. Greater distractor values were associated with higher accuracy only in TMS-MIP condition, but not in non-TMS MIP condition. (**e**) In contrast, the relationship between D-HV and accuracy was comparable between TMS and non-TMS trials when TMS was applied over the control MT region. Shaded regions denote standard error.

The online version of this article includes the following figure supplement(s) for figure 3:

**Figure supplement 1.** An alternative approach of testing the MIP-TMS effect involved collapsing data from all participants and analysing them in a single mixed-effects model.

**Figure supplement 2.** It is noticeable that participants may evaluate choice attributes in a non-linear manner.

**Figure supplement 3.** When TMS was applied to MIP it increased the effect of distractors (indexed by D-HV) on both accuracy and RT, although there were some differences in the precise manner in which accuracy and RT effects manifested.

**Figure supplement 4.** Another way to look at the divisive normalisation effect is to consider effects in a different GLM like that shown in *Figure 2—figure supplement 1*.

The results of GLM3 confirmed once again that in the control data, on average, higher accuracy was associated with larger HV-LV differences (*t*(30) = 17.04, p<0.001, *d*=3.06, CI = [0.76, 0.97]; *Figure 3a*) and lower HV +LV sum values (*t*(30) = –3.50, p=0.001, *d*=0.63, CI = [-0.33,–0.09]; *Figure 3b*). The average impact of D-HV on accuracy was not significant (*t*(30) = 0.69, p=0.496, *d*=–0.12, CI = [–0.08, 0.16]; *Figure 3c*), which, as already explained above, is consistent with the simultaneous presence of both positive and negative distractor effects on the hard and easy trials respectively that we have previously demonstrated (*Figure 2d*). Next, a Site (MIP/MT) x Stimulation (TMS/Non-TMS) x Distractor Location (contralateral/ipsilateral) ANOVA was applied to the beta values of each predictor (*Figure 3*). We focused on examining the three-way interaction relating to the distractor value, in which a significant effect would suggest a robust *TMS effect* when it was applied to a *specific brain region* and when the distractor was presented at a *specific location*. When examining three-way interactions of this type, we found no significant effects associated with predictors that did not incorporate the distractor value D such as the HV-LV predictor (F(1,30) = 1.50, p=0.230, $\eta_p2$ p20.05; *Figure 3a*; *Table 1*) or the HV +LV predictor (F(1,30) = 3.06, p=0.090, $\eta_p2$ p20.09; *Figure 3b*). This suggests that these interactions were

**Table 1.** F- and p-values associated with a Site (MIP/MT) x Stimulation (TMS/Non-TMS) x Distractor Location (contralateral/ipsilateral) ANOVA applied to the beta values of each regressor in GLM3 (HV +LV, HV-LV, D-HV), when GLM3 predicts choice accuracy.

| | HV +LV | | HV-LV | | D-HV | |
|---|---|---|---|---|---|---|
| | *F*-value | p-value | *F*-value | p-value | *F*-value | p-value |
| Site | 1.01 | 0.324 | 0.51 | 0.482 | 0.24 | 0.625 |
| Stimulation | 3.00 | 0.094 | 0.69 | 0.412 | 0.87 | 0.358 |
| Distractor Location | 0.55 | 0.462 | 0.85 | 0.365 | 0.61 | 0.442 |
| Site * Stimulation | 0.06 | 0.812 | 1.65 | 0.209 | 0.40 | 0.533 |
| Site * Distractor Location | 0.35 | 0.560 | 3.01 | 0.093 | 0.54 | 0.468 |
| Stimulation * Distractor Location | *1.65* | *0.209* | *0.40* | *0.531* | *0.08* | *0.786* |
| Site * Stimulation * Distractor Location | *3.06* | *0.090* | *1.50* | *0.230* | *4.44* | *0.044\** |

*\*p < 0.05.*

unaffected by TMS. Critically, however, the D-HV predictor showed a statistically significant three-way interaction (Site x Stimulation x Distractor Location: $F(1,30) = 4.44$, p=0.044, $\eta_p^2$ p2 0.13; *Figure 3c*). This is consistent with a relative increase in the positive distractor effect (previously associated with vmPFC; *Chau et al., 2014*; *Fouragnan et al., 2019*) at the expense of the divisive normalisation effect associated with intraparietal areas such as MIP. No other main or two-way interaction effects of D-HV were observed in the ANOVAs, $F<0.87$, $p>0.358$ (*Table 1*).

In order to explore the three-way interaction on the D-HV predictor, we split the data into contralateral and ipsilateral sets, that is trials in which the distractor was presented contralaterally/ipsilaterally to the TMS pulse, and performed a Site x Stimulation ANOVA on each set of trials. In each ANOVA, the terms associated with the opposite side were also entered as covariates. We found no Site x Stimulation effects in the ipsilateral data set ($F<1.87$, $p>0.180$). Since traditional frequentist statistics are less ideal for supporting claims of null effect, we performed Bayesian tests to compare the D-HV effect between TMS and Non-TMS trials in the ipsilateral data set (when the distractor had been presented ipsilateral to the MIP TMS). The results confirmed that there was an absence of TMS effect when it was applied over ipsilateral MIP (BF10=0.209) or MT (BF10=0.271). In the contralateral data, we found a significant Site x Stimulation interaction effect, $F(1,26) = 4.99$, p=0.034, $\eta_p^2$ p2 0.16 (no other effects were significant, $F<0.73$, $p>0.400$), indicating that the Site x Stimulation x Distractor Location effect was driven by the contralateral presentation condition (when the distractor was presented contralateral to the MIP TMS). In other words, this is consistent with a relative increase in the positive distractor effect (previously associated with vmPFC) at the expense of the divisive normalisation effect associated with intraparietal areas such as MIP that occurs mainly when distractors are presented contralateral to the TMS site.

To clarify the nature of this effect, we split the contralateral data further into MT and MIP sets and repeated the ANOVA, entering only the Stimulation factor and including all other conditions as covariates, on each set. We found a significant effect of TMS on MIP conditions ($F(1,24) = 4.32$, p=0.049, $\eta_p^2$ p2 0.15), with TMS trials showing a more positive D-HV effect than Non-TMS trials. The MIP-TMS effect became even clearer after the grey matter volume (GM) of the same region was also entered as a covariate ($F(1,23) = 7.02$, p=0.014, $\eta_p^2$ p2 0.23; the next section explains the importance of considering the GM and explains how the GM indices were obtained). In contrast, we found no effect in MT conditions ($F(1,24) = 1.24$, p=0.277, $\eta_p^2$ p2 0.05), and this lack of TMS effect was confirmed by an additional Bayesian test (BF10=0.225). The results remained similar even after entering the GM of MT as an additional covariate ($F(1,23) = 0.19$, p=0.664, $\eta_p^2$ p2 0.01). These results suggest that TMS of MIP had a significant impact on promoting the positive distractor effect on accuracy at the expense of the opposing negative (divisive normalisation) distractor effect dependent on intraparietal sulcus areas such as MIP (*Chau et al., 2014*; *Louie et al., 2011*). The effect was especially clear when the distractor was presented contralaterally to the MIP TMS site. Finally, the MIP-TMS effect was even more robust when we analysed the data from all participants together in a mixed-effects model (*Figure 3—figure*

*supplement 1*) or when we considered that participants may evaluate choice attributes in a non-linear manner (*Figure 3—figure supplement 2*).

While our primary focus is on accuracy as an index of response selection, our diffusion model22, like most diffusion models, suggests that in many cases, factors that increase response selection accuracy will also increase response selection RT. This was true in the present case (*Figure 2—figure supplement 1*).

Finally, as explained in *Figure 2—figure supplement 1*, the negative distractor effect should have become smaller when HV +LV was large (on such trials the distractor constitutes a smaller part of the total value of the stimuli and ultimately it is this total value that determines divisive normalisation). This was revealed as a positive (HV +LV)D interaction effect. In *Figure 3—figure supplement 4*, we showed the (HV +LV)D effect also became marginally less positive after MIP-TMS (F(1,30) = 3.77, p=0.062, $\eta_p$2p20.112).

In summary, the D-HV term indexes an important aspect of the influence of the distractor on behaviour. At baseline it is associated with two significant absolute effects; large distractor values are associated with higher accuracy when decisions are difficult (*Figure 2d*, left) and they are associated with lower accuracy when decisions are easy (*Figure 2d*, right). Thus, the balance of the distractor effect changes across the decision space defined by the choice values (such as the differences in their values). The balance of distractor effects also significantly changes with the disruption of MIP using TMS; the positive distractor effect becomes stronger at the expense of the negative distractor effect (*Figure 3c*).

## MRI data: VBM confirms link between MIP and the impact of TMS on the distractor effect

So far, we have provided evidence that MIP is causally related to the negative distractor effect because the antagonistic positive distractor effect emerged prominently and to a significantly greater extent after MIP was disrupted. To investigate the importance of MIP, and the impact of its disruption further, we sought an explanation of individual variation in effects. We might expect individual variation in MIP volume to be related to the degree of TMS modulation of the distractor effect. Recent studies suggested that those with smaller GM in the target region also demonstrate stronger behavioural changes after receiving TMS (*Ye et al., 2019*). Thus, in individuals with larger MIPs, the negative distractor effect should be less susceptible to TMS disruption and the opposing positive distractor effect should appear weaker.

To test this, we performed a voxel-based morphometry (VBM) analysis to examine the relationship between GM and the TMS effect on participants' decision-making. The analysis was focused on parietal and occipital cortex in the same hemisphere to which TMS had been applied. The same GLM3 (HV-LV, HV +LV, D-HV) as described in the behavioural analysis was used. In order to extract the effects of TMS, we subtracted the beta values associated with Non-TMS trials from those associated with TMS trials (in contralateral conditions). This was conducted once for MT conditions, and once for MIP conditions. Interestingly, the effect of MIP TMS on the distractor value's (D-HV) impact on decision accuracy showed a negative relationship with MIP GM (p=0.029, TFCE corrected, centred around MNI X(–30), Y(–52), Z(42), *Figure 4a*). This implies that in individuals with larger MIP GMs, there was less difference between TMS and Non-TMS trials (less relative increase in the positive distractor effect at the expense of the negative distractor effect), because TMS had a weaker impact on disrupting the MIP-related negative distractor effect. No significant GM differences were observed in any other parietal and occipital regions (p>0.170).

We followed these tests up by extracting the GM at MIP and MT. First, we illustrate the VBM results again by showing that there was a significant partial correlation between MIP GM and MIP TMS impact on the D-HV effect (*r*(27) = –0.61, p<0.001; *Figure 4b*, left panel), after controlling for the other explanatory variables entered in the VBM (i.e. TMS effects on HV-LV and HV +LV). In addition, the correlation remained significant even without controlling for HV-LV and HV +LV (*r*(29) = –0.37, p=0.043). Since visual inspection revealed a potential outlier with small MIP GM and a large TMS effect on the D-HV predictor, we repeated the partial correlation analysis by excluding this data point. The results did not change qualitatively (*r*(26) = –0.55, p=0.002).

Next, we ran three additional analyses to demonstrate that the correlation was specific to the MIP GM only when TMS was applied to MIP itself (but not when TMS was applied to the control MT

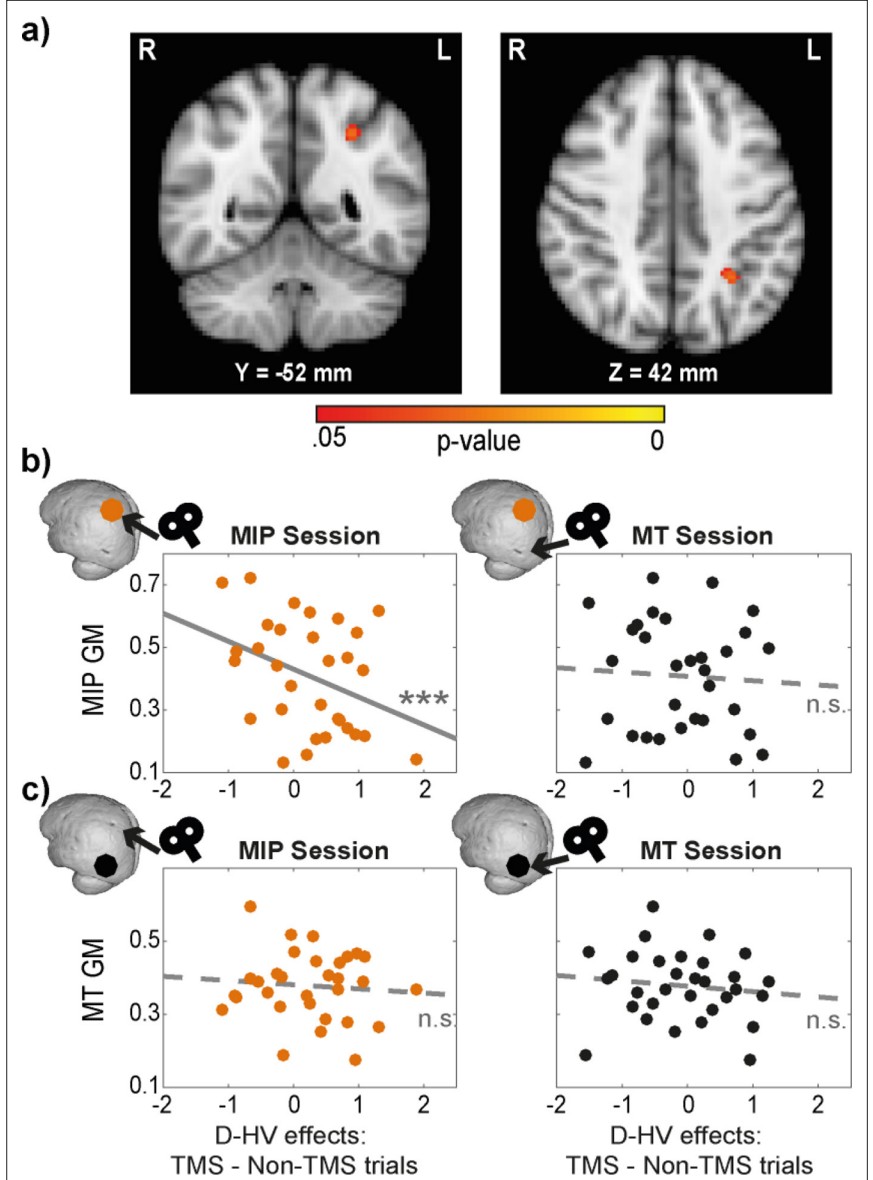

**Figure 4.** Distractor effects of individuals with larger MIPs were less susceptible to TMS. (**a**) Statistical map displayed on a standard MNI brain shows a significant relationship between TMS effect (difference in distractor effect between TMS and Non-TMS trials) and MIP. (**b**) Local grey matter volume (GM) plotted as a function of TMS effect. (Left) An illustration of the effect in (**a**). Individuals with larger MIPs also revealed less positive change in the D-HV effect between TMS and Non-TMS trials in MIP sessions. This suggested that the MIP-related negative distractor effect was less disrupted by TMS when MIP was larger. (Right) In contrast, the difference in D-HV effect between TMS and Non-TMS trials in MT sessions was unrelated to the volume of the MIP region. (**c**) A similar analysis was performed for the MT GM. Both MIP (Left) and MT (Right) TMS effects (difference in D-HV effect between TMS and Non-TMS trials) were unrelated to the MT GM. Top left inserts show the approximate location of the GM (coloured dot) and the site of the TMS effect (arrow) used in each correlation. *** p<0.001.

The online version of this article includes the following figure supplement(s) for figure 4:

**Figure supplement 1.** VBM reveals association between MIP TMS and frontal region.

region). First, there was no correlation between MIP GM and TMS effect when it was estimated in the control sessions where MT was stimulated ($r(27) = -0.05$, p=0.816; *Figure 4b* right). Second, despite finding no effect in MT in the VBM analysis, we extracted the GM in the control MT region and tested whether it was related to the TMS effect in the experimental MIP session. We found no significant correlation ($r(27) = -0.19$, p=0.325; *Figure 4c* left) even though this analysis is less conservative than

the VBM analysis illustrated in *Figure 4a*. Finally, there was also no relationship between the GM of the control MT region and TMS effect extracted from the control MT-TMS sessions ($r(27) = -0.03$, p=0.871; *Figure 4c* right). The conclusion that there was a null effect was supported by supplementary Bayesian analyses (BF10 <0.275 in all three relationships illustrated in *Figure 4b* right and 4 c left and right).

Finally, we note that the correlation between MIP GM in each individual and the impact of MIP TMS on the D effect in behaviour was more strongly negative than the correlation between MIP GM in each individual and the impact of MT TMS on the D effect in behaviour (i.e. comparing the correlations in *Figure 4b* left and right; $z=-2.47$, p=0.014).

## Eye-tracking data: TMS affects gaze shifts between D and HV

Previous work has suggested that the positive distractor effect, which is prevalent on hard trials (*Figure 2d*) and which becomes more prominent when MIP is disrupted (*Figure 3c*), is linked to particular patterns of eye movement. *Chau et al., 2020* showed that there is a positive correlation between distractor value and gaze shift between the D and HV options. As such, larger distractor values are associated with more D-and-HV gaze shifts and, ultimately, more accurate HV choices are made. This suggests that accumulation of evidence in favour of the HV, as opposed to the LV, option is prolonged when D captures overt attention, and this eventually leads to more accurate decision-making. Similarly, in other settings, participants who are allowed extra time to revise their initial decisions tend, ultimately, to make more accurate decisions (*Resulaj et al., 2009*; *van den Berg et al., 2016*). Therefore, we tested whether the positive relationship between distractor value and D-to-HV gaze shift was replicable in the current study. We also tested whether the positive relationship became even stronger after the TMS disrupted the negative distractor effect and spared the positive distractor effect. Hence, we again applied GLM3 (HV-LV, HV +LV, D-HV) to predict gaze shifts between D and HV.

First, we showed that larger D-HV values were related to more gaze shifts between D and HV ($t(30) = 4.75$, p<0.001, $d=0.85$, CI = [0.03, 0.08]). The result is similar to that reported by Chau and colleagues (*Chau et al., 2020*). However, in the previous study the relationship between the difference in D/HV values, D-HV, and gaze shifts was apparent for D-to-HV gaze shifts whereas in the present analysis the association was with gaze shifts between HV and D in either direction. In addition, HV-LV and HV +LV had no clearly significant effect on the bidirectional gaze shifts ($t(30) < 1.55$, p>0.131). We then performed a critical test that examined whether large distractor values were more strongly related to more gaze shifts between D and HV after MIP was disrupted using TMS. This was done by comparing the effect of D-HV on gaze shifts between HV and D using a Site (MIP/MT) by Side (contralateral/ipsilateral) by Stimulation (TMS/Non-TMS) ANOVA. This was analogous to the analysis in *Figure 3c* that examined how TMS modulated the distractor's influence on choice behaviour. Note that five participants were excluded in this analysis due to the absence of gaze shifts between HV and D in some conditions of this ANOVA. The results showed a significant Site ×Stimulation interaction effect, $F(1,25) = 8.85$, p=0.006, $\eta_p2$p20.26. Follow-up ANOVAs with factors of Side and Stimulation revealed in MIP sessions the effect of distractor value on gaze shifts was significantly higher in TMS compared to Non-TMS trials ($F(1,25) = 5.30$, p=0.030, $\eta_p2$p20.18), while in MT sessions there was no significant difference between TMS and Non-TMS trials ($F(1,25) = 1.62$, p=0.214, $\eta_p2$p20.06).

To avoid excluding data from five participants due to splitting the data multiple times into small sets, we ran a similar analysis by collapsing the conditions of Side (contralateral/ipsilateral). Then we performed a two-way Site by Stimulation ANOVA to compare the effect of D-HV on gaze shift between HV and D. Again, the results showed a significant two-way interaction effect, $F(1,30) = 6.73$, p=0.014, $\eta_p2$p20.18 (*Figure 5a*, right; *Table 2*). No effects were found when a similar ANOVA was performed to compare the effect of HV-LV and HV +LV on the gaze shifts between HV and D ($F<2.71$, p>0.110; *Figure 5a*, left and middle, respectively).

The impact of the distractor value, D, on gaze shifts is sometimes more apparent when D-to-HV unidirectional gaze shifts are considered (as opposed to gaze shifts in either direction) (*Chau et al., 2020*). We examined these gaze shifts separately from those in the opposite direction (HV-to-D shifts). When we performed a two-way ANOVA, a Site by Stimulation interaction was only found in the D-to-HV direction ($F(1,30) = 5.82$, p=0.022, $\eta_p2$p20.16; *Figure 5b*, right), but not the HV-to-D direction ($F(1,30) = 1.03$, p=0.319, $\eta_p2$p20.03; *Figure 5c*, right). These results show that distractors with higher

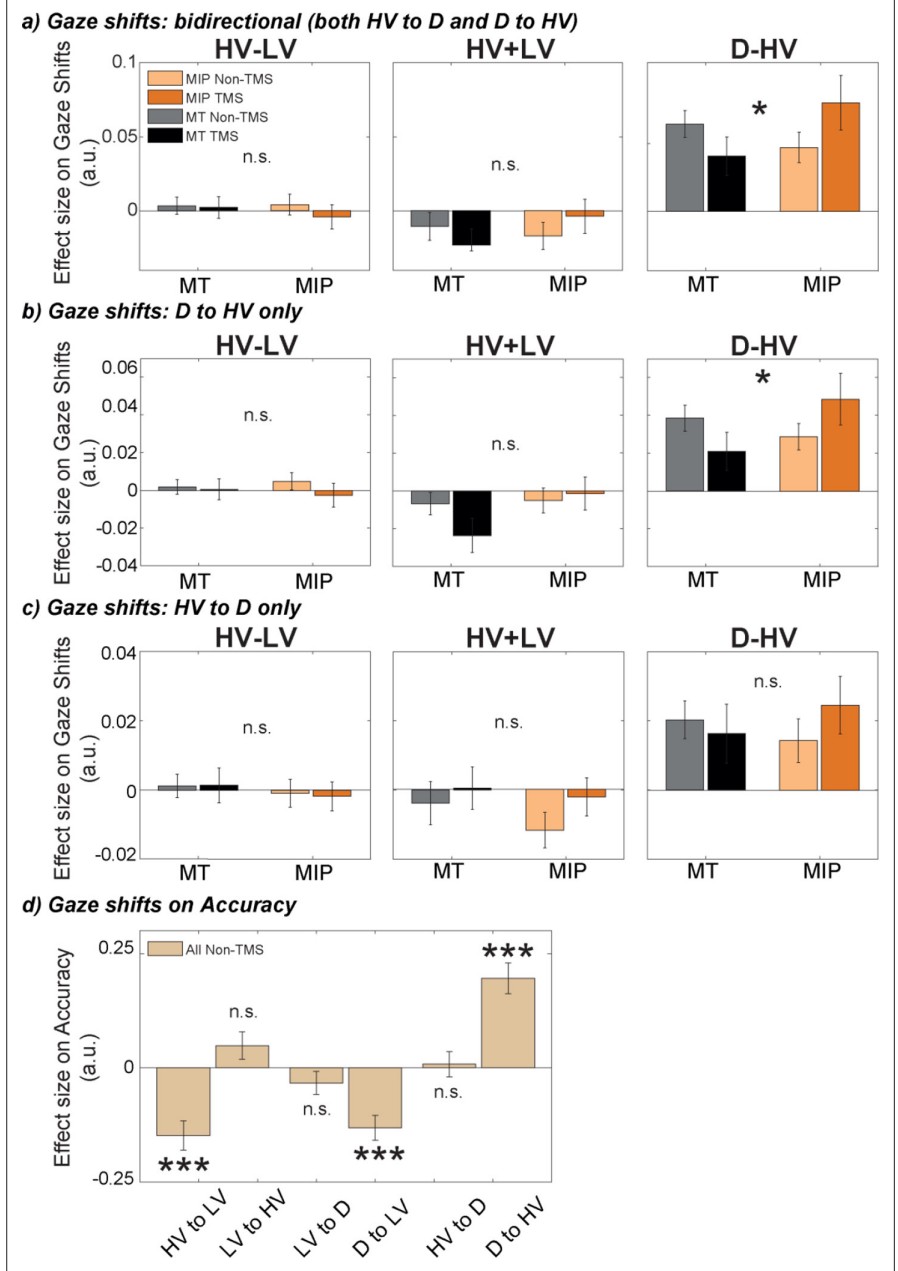

**Figure 5.** The positive distractor effect was more strongly related to eye movements, especially between the distractor D and HV, after MIP was disrupted by TMS. GLM3 was used to predict gaze shifts between different options, namely (**a**) bidirectional gaze shifts between D and HV, (**b**) directional gaze shift from D to HV, and (**c**) directional gaze shifts from HV to D, for each participant and each condition. TMS over MIP increased the number of gaze shift from D to HV (Site x Stimulation interaction, * *P*<0.050), but not from HV to D. (**d**) GLM4 was used to test the impact of gaze shifts between all possible options on accuracy. Shifts towards the low-value option were found to decrease accuracy, while shifts from D to HV increased accuracy (*** p<0.001). Error bars denote standard error.

The online version of this article includes the following figure supplement(s) for figure 5:

**Figure supplement 1.** Eye-movement patterns show shift under MIP stimulation.

values lead to increased numbers of gaze shifts from the distractor to the high-value option, and importantly, that this effect is increased by applying TMS over MIP.

In the analyses above, we used the predictors HV-LV, HV +LV, and D-HV to predict gaze shifts in order to be consistent with the previous analyses of the choices participants took (*Figure 3*). However,

**Table 2.** *F*- and p-values associated with a Site (MIP/MT) x Stimulation (TMS/Non-TMS) ANOVA applied to the beta values of each regressor in GLM3 (HV +LV, HV-LV, D-HV), when GLM3 predicts gaze shifts between D and HV.

| | HV +LV | | HV-LV | | D-HV | |
|---|---|---|---|---|---|---|
| | *F*-value | p-value | *F*-value | p-value | *F*-value | p-value |
| Site | 0.41 | 0.526 | 0.11 | 0.739 | 0.46 | 0.503 |
| Stimulation | 0.00 | 0.965 | 0.62 | 0.437 | 0.15 | 0.698 |
| Site * Stimulation | 2.71 | 0.110 | 0.43 | 0.515 | 6.73 | 0.014* |

*p<0.05.

in order to predict gaze shifts, it might be argued that the effects of individual option values, instead of differences in, or sums of, values of options are more easily interpretable. We therefore repeated the same analyses, but using the predictors HV, LV, and D, instead of HV-LV, HV +LV, and D-HV. The results did not change qualitatively (*Figure 5—figure supplement 1*).

Finally, to explore the effects these gaze shifts had on decision-making, we used gaze shifts in all possible directions (HV to LV, LV to HV, LV to D, D to LV, HV to D, and D to HV; *Figure 5d*) to predict decision accuracy. We found that increased numbers of gaze shifts towards the low-value option were associated with lower accuracy (HV to LV: *t*(26) = –4.6, p<0.001, *d*=–0.88, CI = [-0.21,–0.08]; D to LV: *t*(26) = –4.74, p<0.001, *d*=–0.91, CI = [-0.19,–0.07]). Importantly, we found increased accuracy to be associated with increased gaze shifts from the distractor to the high-value option (D to HV: *t*(26) = 5.77, p<0.001, *d*=1.11, CI = [0.13, 0.27]). No other gaze shifts revealed significant results (p>0.116). Together, these findings suggest that after TMS disrupts the negative distractor effect mediated by MIP, large distractor values are more influential in promoting D-to-HV gaze shifts which, ultimately, are associated with more accurate choices.

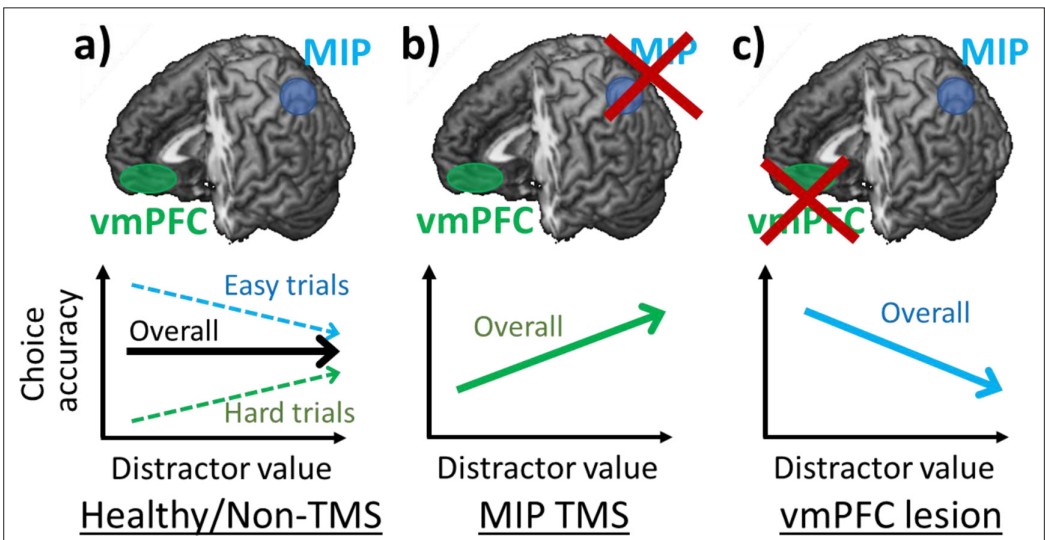

**Figure 6.** Positive and negative distractor effects co-exist and are causally related to frontal and parietal regions respectively. (**a**) People's choices are influenced by the presence of seemingly irrelevant distractors and the precise effects vary across the decision space. Choices are improved by large distractor values when they are difficult (positive distractor effect; green arrow). In contrast, choices are impaired by large distractor values when they are easy (negative distractor effect; blue arrow). Previous studies showed that positive and negative distractor effects are related to vmPFC and MIP respectively. (**b**) When MIP is disrupted using TMS, the negative distractor effect is reduced, sparing the opposing positive distractor effect. (**c**) In contrast, in patients with vmPFC lesions, the positive distractor effect is reduced, sparing the opposing negative distractor effect.

## Discussion

Several lines of evidence suggest that decisions can be altered by adding seemingly irrelevant distractors to decision-making tasks. Two contrasting effects have been reported; the relative value of a distractor has been associated with both improved (positive distractor effect) and impaired (negative distractor effect) decision accuracy (*Chau et al., 2014*; *Louie et al., 2013*). At first glance, these effects appear mutually exclusive and it has recently been argued that neither effect is present (*Gluth et al., 2018*; *Gluth et al., 2020*). However, it has been demonstrated that both effects co-exist and vary across individuals (*Chau et al., 2020*; *Webb et al., 2020*). Data obtained from humans and monkeys also suggest the possibility that the two distractor effects are associated with the operation of different brain regions. The negative distractor effect predicted by divisive normalisation models is associated with intraparietal sulcal regions, such as MIP and LIP (*Chau et al., 2014*; *Louie et al., 2013*; *Louie et al., 2011*), while the positive distractor effect has been linked with vmPFC (*Chau et al., 2014*; *Fouragnan et al., 2019*). Moreover, it has been suggested not only that both distractor effects co-exist, but also that each effect prevails in different parts of decision space (*Figure 6a*). Specifically, the negative distractor effect is most prominent in decisions that are easy or with options that are generally poor; whereas the positive distractor effect is most prominent in decisions that are difficult or with options that are generally valuable (*Figure 2c and d* and *Figure 2—figure supplement 1*). These variations in the distractor effect are captured by a dual-route model that incorporates both a divisive normalisation mechanism and a mutual inhibition mechanism in separate decision routes that compete in a parallel race (*Chau et al., 2020*). It is clear that multiple brain mechanisms are concerned with choice selection and that these span not just parietal cortex and vmPFC but also include orbitofrontal cortex, premotor cortex, and cingulate cortex as well as subcortical regions (*Rushworth et al., 2012*; *Kolling et al., 2016*; *Hunt and Hayden, 2017*; *Chau et al., 2018*). What determines which mechanism predominates in a given setting is likely to be related to the speed with which it operates in a given situation and with a given input (*Chau et al., 2014*) and the certainty of the evaluations that it makes. The current results again suggest an association between these different routes and different brain regions.

In the current study, we demonstrated that after disrupting the parietal cortex using TMS, the negative distractor effect was reduced and so the remaining positive distractor effect emerged more robustly (*Figures 3c and 6b*). Additionally, an analysis of structural differences in parietal regions provided further evidence for the role of MIP. Individuals with larger MIPs were less susceptible to the TMS disruption and showed less impairment in their negative distractor effects (*Figure 4*). These results obtained using TMS and structural scans are in line with results of functional scans showing that MIP signals were weakened by large distractor values presented on the ipsilateral side (*Chau et al., 2014*). Notably, we found that the MIP region in which grey matter variance predicted variance in the size of MIP TMS effects was slightly anterior to the MIP region targeted with TMS. This may reflect the fact that the impact of individual variance in MIP is most apparent at its edges rather than at its centre. No similar relationship was apparent between the grey matter in any other parietal, occipital, or temporal area in the stimulated hemisphere. A whole brain analysis found only one other region in perigenual anterior cingulate cortex (pgACC; *Figure 4—figure supplement 1*).

While we initially sought evidence of any positive relationship (i.e. larger MIP GM associated with stronger negative distractor effects, the disruption of which would have a stronger effect overall), we found no evidence for any such pattern. This may be consistent with a recent report that relationships between individual variation in brain structure and individual variation in behaviour are not strong (*Kharabian Masouleh et al., 2019*). It might, however, be reasoned that it ought to be difficult to relate variation in behaviour per se to variation in brain structure because a behaviour reflects an aggregate, net output of multiple processes distributed across numerous brain areas. It should, however, be easier to relate variation in TMS effects on behaviour to variation in brain structure because the variation in TMS effects on behaviour are mediated by one specific brain region. The negative relationship that was found is most simply interpreted as suggesting that lower MIP GM makes MIP more easily and reliably disrupted by TMS. Although this result does not speak to the relationship between MIP GM and the negative distractor effect per se, it nevertheless informs us of the locus of the TMS effect and confirms that MIP plays a crucial role in the expression of the negative distractor effect. We note, however, that even if individual variation in MIP structure does

not predict individual variation in susceptibility to divisive normalisation, individual variation in MIP activity patterns do predict individual variation in similar behavioural effects (*Chau et al., 2014*).

In contrast, a causal relation between the vmPFC and a positive distractor effect has been observed in both humans and monkeys (*Noonan et al., 2010*; *Noonan et al., 2017*). In both species, individuals with vmPFC lesions, unlike control or pre-operative individuals, were less accurate in their decisions when high-value distractors were presented. In vmPFC lesion individuals, the manifestation of a negative distractor effect is possibly due to the reduction of the opposing positive distractor effect (*Figure 6c*). Collectively, these findings demonstrate that positive and negative distractor effects are causally related to the neural processes in vmPFC and parietal cortex respectively.

It has been argued that the positive distractor effect, which is linked to vmPFC, is mediated by overt attention. According to the mutual inhibition model, the presence of an appealing distractor can boost the overall inhibition in a neural network and avoid stochastic choices. Indeed, similar predictions are apparent in other sequential sampling models that explain changes of mind in decision-making (*Resulaj et al., 2009*; *van den Berg et al., 2016*). These models suggest that the evidence accumulation process, which guides decision-making, continues even after an initial decision is made. When people are allowed to revise their choice after a brief delay, they tend to make more accurate decisions afterwards. All these models share the prediction that a distractor is able to delay the decision-making process, causing more accurate decisions. Indeed, eye movement data support this notion. In essence, large value distractors should attract more gazes and delay the choices between the two alternative options. Interestingly, when the distractor is perceived as unavailable, attention is quickly shifted from it to the high-value option (*Chau et al., 2020*). In the current study, we found that these gaze shifts from the distractor to the high-value option were indeed associated with increased accuracy, and crucially, that these gaze shifts were promoted when the positive distractor effect became more prominent after MIP TMS (*Figure 5b*).

It is important to note that a positive distractor effect can be induced in some regions of distractor space in divisive normalisation-based models (*Louie et al., 2013*; *Webb et al., 2021*). This occurs because, as a noisy representation of the distractor increases in value, it is disproportionately more likely to be chosen than the LV option as opposed to the HV option. Whether this effect exerts a large influence in the current study is unclear given that participants are instructed never to choose the distractor. It is, nevertheless, the case that the distractor is, albeit infrequently, chosen by participants. Despite their differences, it is notable across a range of models that it is the operation of a choice comparison process in the context of noise that makes it possible to account for positive distractor effects. This is true whether it is a recurrent neural network model emphasizing mutual inhibition between pools of neurons representing each choice (*Chau et al., 2014*), diffusion models of evidence accumulation for choice selection (*Chau et al., 2020*), or even in certain parts of decision space in divisive normalisation (*Webb et al., 2021*). While some models place greater emphasis on the presence of noise in the comparison process, other emphasize noise in the representation of the choices themselves. It is possible to envisage models in which the asymmetric impact of a noisy distractor representation on rejection of the LV option is greatest when HV and LV options are far apart in value. The empirical findings reported here and elsewhere (*Chau et al., 2014*; *Chau et al., 2020*), however, suggest that the greatest impact occurs when HV and LV are close in value and it is difficult to choose between them. The high value distractor augments and protracts the comparison process during these difficult choices so that HV is more likely to be chosen.

A third account of how the distractor affects decision making is by attentional capture (*Gluth et al., 2018*; *Gluth et al., 2020*). It suggests that overt attention is attracted by the presence of a salient distractor such that inaccurate choices will be made by choosing the distractor itself. It is noticeable that these three accounts (positive distractor effect, negative distractor effect, and attentional capture) are not mutually exclusive, but instead they can all co-exist and each account describes one aspect of how the distractor influences decision making (*Figure 2d* and *Figure 2—figure supplement 2*).

There are two limitations in the current study. First, we did not directly test whether the positive distractor effect is generated in the vmPFC. Second, an alternative explanation for the presence of a positive distractor effect after MIP-TMS is that it is due to a direct change in the divisive normalisation computation in the MIP itself. We argue, however, that another interpretation is that by perturbing the MIP and reducing its associated negative distractor effect then this allows for stronger expression of the positive distractor effect associated with the vmPFC. This link is based on findings of previous

studies that suggest a relationship between the positive distractor effect and vmPFC (*Chau et al., 2014*) and the greater prominence of negative distractor effects after vmPFC is damaged (*Noonan et al., 2010*; *Noonan et al., 2017*). A future study which includes stimulation of the vmPFC could examine this interpretation further. Note however, that TMS over frontal regions can cause discomfort in participants and would require an adjustment of the stimulation protocol. Perhaps most critically, it may not even be possible for TMS effects to be induced at the depth below the scalp at which vmPFC lies. Additionally, it is important to note that the effects in the VBM analysis that we do report are relatively small (*Figure 4* and *Figure 4—figure supplement 1*). However, we argue that several independent effects can be identified – changes in behavioural performance, changes in eye-movement patterns, and grey matter differences – and they all provide converging evidence for the occurrence of positive and negative distractor effect as well as the role of MIP in decision-making.

The current study focused on the interactions that occurred between available options and the distractor as a function of their overall value and the involvement of the MIP in mediating the impact of the distractor. Recent model developments have suggested that in multi-attribute choices, the interactions between options could occur at the level of the representations of their individual, component attributes, prior to integration into an overall value. These models include the use of divisive normalisation (*Landry and Webb, 2021*; *Dumbalska et al., 2020*) or leaky-competing-accumulator model (*Usher and McClelland, 2004*). Although the current study makes no specific claim about whether MIP is causally related to the divisive normalisation mechanism at the level of overall value or individual attribute, it will be important to test these two alternative hypotheses in future studies.

In summary, we show that TMS-induced disruption of MIP increases the positive distractor effect, i.e. the relationship between increased distractor values and improved accuracy. We argue that, in the current study, positive and negative distractor effects co-occur, and that the disruption of one gives way to an increased expression of the other.

## Methods

### Participants

A total of 31 neurotypical, right-handed participants (17 female) with a mean age of 21.90 (SD = 3.08) were recruited. Previous studies that showed a behavioural effect after rTMS was applied to intraparietal area involved sample sizes that ranged between 10 and 15 participants (*Gould et al., 2012*; *Coutlee et al., 2016*; *Dormal et al., 2012*; *Hayashi et al., 2013*) and the sample size of the current study surpassed this range. Prior to the experiment, each participant received information about the procedures and completed a screening questionnaire to ensure the safe use of TMS and magnetic resonance imaging (MRI). Each participant was paid between HKD 400 and HKD 500, depending on task performance (see *Decision-Making Task*). The experiment was approved by the Human Subjects Ethics Committee of the Hong Kong Polytechnic University and informed consent was obtained from all participants (HSEAR20151208001).

### Procedures

Participants took part in a total of four experimental sessions. In Session 1, experimental procedures were explained to the participants and all safety screening was completed to ensure that participants were suitable to be involved. Then, we established their individual motor threshold using TMS (see *Transcranial Magnetic Stimulation*) and asked them to practice the decision-making task. In Session 2, participants' structural brain scans were obtained using MRI (see *MRI data acquisition*). Sessions 3 and 4 were experimental sessions in which they completed the decision-making task while their eye-movements were tracked (see *Eye movement recording and preprocessing*) and repetitive transcranial magnetic stimulation (rTMS) was applied. In Sessions 3 and 4, the rTMS was applied over either their MIP or a closely adjacent control brain region – the MT region. The order of MIP/MT sessions was randomised across participants. Each experimental session took approximately 60 min. Sessions 3 and 4 were scheduled to be a minimum of seven days apart.

### Decision-making task

Participants completed a multi-alternative value-based decision-making task that was similar to that of previous studies (*Chau et al., 2014*; *Chau et al., 2020*; *Figure 1a*). In each trial, three coloured

rectangles were presented and participants were asked to choose the best option. The rectangles' value was defined by their colour and orientation, indicating the associated reward magnitude and probability respectively. Specifically, each stimulus was associated with one of six reward magnitudes ($25, $50, $75, $100, $125, $150), indicated by colour, ranging from blue to green and one of seven probabilities (1/8, 2/8, 3/8, 4/8, 5/8, 6/8, 7/8) indicated by orientation, ranging from 0° to 90° (*Figure 1b*). The direction of the mapping between visual (colour, orientation) and decision (magnitude, probability) properties were randomised across participants.

At the beginning of each trial, participants were presented with a fixation cross at the centre of the screen (2.5 s), followed by the stimulus onset, during which three options represented by rectangular bars were shown in randomly selected screen quadrants. On 33% of the trials, after 50ms five TMS pulses were applied at a frequency of 10 Hz. After another 50ms, coloured boxes were presented around each stimulus, indicating which two options were available for choice (orange boxes), and which option was the unchooseable distractor (purple box). Participants were instructed to indicate their choice within 1.5 s from the onset of this decision phase. To indicate their decision, participants pressed one of four keys ('f', 'v', 'n', 'j') on a keyboard, associated with the four screen quadrants (top left, bottom left, bottom right, top right respectively). After a stimulus was chosen, the box surrounding it turned red (0.5 s). Then the edge of each stimulus was coloured either gold or grey to indicate which stimuli were rewarded (1 s). If the distractor or the empty quadrant were chosen, or no response was provided before the 1.5 s deadline, feedback ('Error!' or 'Too slow!' respectively) was provided.

In order to ensure that all stimuli were attended to, a matching task was added to 1/6 of trials. In these trials, participants were presented with the word 'MATCH' (1 s), followed by a rectangular stimulus (2 s) in the centre of the screen after they indicated their decision. The participants were asked to identify in which screen quadrant of the current trial this stimulus was presented and indicate their choice in the same way as described above. If the stimulus did not match any of the three stimuli of the current trial, the participant was asked to select the blank quadrant. Feedback for the matching task ('Correct Match'/'Incorrect Match') was provided in gold/grey writing (1.5 s), before the main decision-making trial continued. Unless stated otherwise, matching trials were not included in the analysis.

In each experimental session (Session 3 or 4), participants completed 30 practice trials, followed by three blocks of 90 experimental trials. The task was written using Presentation software (Neurobehavioral Systems Inc, CA, USA), run on a PC, and presented on a 23-inch LCD monitor operating at a refresh rate of 60 Hz and a resolution of 1,920×1,080 pixels.

## Transcranial magnetic stimulation:

TMS was applied using a MagVenture MagPro X100 biphasic stimulator (MagVenture, Denmark) via a figure-of-eight coil (coil winding diameter 75 mm, MagVenture C-B60). Participants were asked to place their heads onto a chin rest to minimise head movements. The position of the coil was tracked using a neuronavigation system (TMS Navigator, Localite, Germany). Each participant's head and their individual MRI were coregistered (during the threshold detection prior to the acquisition of the MRI scan), by marking a number of anatomical landmarks on both the participant's head and on their structural MRI scans (nasion, inion, left/right preauricular point, left/right exocanthion) using a digitising pen. Coregistration was deemed successful when the root mean squared error of the fitting procedure was 5 mm or less.

### Motor Threshold Detection

The stimulation intensity during Sessions 3 and 4 was defined as 100% of the resting motor threshold. During Session 1, the motor threshold was established for each participant and defined as the minimal intensity required to elicit a motor evoked potential (MEP) with a peak-to-peak amplitude of ~50 μV in 50% of stimulations. To identify this intensity, single-pulse TMS was applied to the right primary motor cortex via a coil positioned tangentially to the scalp with the handle pointing backward. MEPs were measured using surface electromyography electrodes placed over the left first dorsal interosseous in a belly-tendon montage. The exact position of the coil was determined individually based on where the largest MEPs could be reliably evoked. This resulted in a resting motor threshold of, on average, 55.94% (SD = 5.79) of maximum stimulator output.

## rTMS

On Sessions 3 and 4, rTMS was applied in biphasic bursts consisting of 5 pulses at a frequency of 10 Hz during 33% of trials (with a minimum of 20 s in between bursts), 50ms after stimulus onset. We aimed to stimulate the left medial intraparietal area (MIP) in experimental sessions and the left middle temporal visual area (MT) in active control sessions. MIP was chosen based on the literature suggesting its role in hand movement (*Mars et al., 2011*) and divisive normalisation effects in humans (*Chau et al., 2014*). MT, which is implicated in visual motion perception (*Born and Bradley, 2005*), was chosen as a control site as it was unlikely to be critical for any aspect of the performance of the current decision-making task but it is relatively close to the brain area of interest. The stimulation sites MIP and MT were located at the standard MNI coordinates X=-30, Y=-58, Z=62 and X=-51, Y=-80, Z=6 respectively, which were individualized by transforming the standard coordinates back to individual MRI scans using FMRIB's Software Library (FSL). Adjustment of the stimulation sites was applied in some participants due to minor inaccuracies of the automated transformation using FSL or due to discomfort but the resulting distributions of stimulation sites were tightly clustered over MIP and MT (*Figure 1c*).

## MRI data acquisition and preprocessing

T1-weighted structural images were collected using a Philips Achieva 3.0T scanner, using an MPRAGE sequence (1x1 x 1 mm$^3$ voxel resolution, 240x240 x 200 grid, TR = 7 \ms, TE = 3.2ms, flip angle = 8°). Voxel-based morphometry (VBM) was performed to explore associations between differences in local grey matter volume (GM) and the impact of TMS on the distractor effect. Structural MRI data were analysed using the FSL-VBM protocol (*Douaud et al., 2007*; *Good et al., 2001*; *Smith et al., 2004*). Each participant's scan was brain-extracted, grey matter-segmented, and nonlinearly registered to the MNI 152 standard space (*Andersson et al., 2007*). The resulting images were averaged to create a study specific grey matter template. Native grey matter images were then nonlinearly registered to this template and modulated to correct for local expansion/contraction due to the non-linear component of the spatial transformation. The modulated images were then smoothed with an isotropic Gaussian kernel (sigma = 3 mm), and a voxelwise GLM was applied using permutation-based nonparametric testing (5000 permutations). Correction for multiple comparisons across space was performed using threshold free cluster enhancement (TFCE).

Since we set out to explore the effect of TMS, we restricted the analysis to the regions in which we applied TMS, that is the left MIP as well as the left MT as a control region. A region of interest (ROI) analysis was chosen because (1) we tested the effect of TMS and our hypotheses were therefore focused on the specific TMS sites used, and (2) all participants were sampled from a neurotypical population, suggesting that any structural differences associated with decision-making would be small. The region of interest covered large areas of the left parietal and occipital grey matter regions, and was defined as the left superior parietal lobule, the left angular gyrus, and the left inferior lateral occipital cortex, as defined by the Harvard-Oxford Cortical Structural Atlas (see also *Figure 2—figure supplement 2*).

## Eye movement recording and preprocessing

Eye gaze data was recorded at a sampling rate of 300 Hz using a TX300 video eye tracker (Tobii Technology, Sweden). Each recording was preceded by the default nine-point calibration procedure. Eye-tracking data were analysed in Matlab using custom scripts based on the Velocity-Threshold fixation identification (I-VT) algorithm (*Komogortsev et al., 2010*). First, small gaps (</=75ms in duration) in the raw eye-tracking data were linearly interpolated, and the noise in the resulting data was reduced through a moving median (window size = 3 samples). The eyes' angular velocity was then calculated in 20ms time windows, and fixations were defined as periods in which the velocity stayed below a threshold of 30 degrees per second. Lastly, fixations which are adjacent in both time (</=75ms difference) and angle (</=.5 degree difference) were merged, and short fixations (<60ms) were discarded.

## Statistical analysis
### Behavioural Data Analysis

Based on the expected value of each stimulus, which was defined as the product of its reward magnitude and probability, the two choosable options in each trial were defined as high-value (HV) or

low-value (LV). We refer to the expected value of the distractor stimulus as 'D'. A trial was defined as accurate when the HV option was chosen, and as incorrect when the LV option was chosen. Trials in which the distractor or the empty quadrant were chosen, and trials in which no response was given (4.13%) were discarded. We applied three generalised linear models (GLMs) with a binomial data model (applied using the Matlab function 'glmfit') to predict each participants' accuracy:

GLM1: $\beta_0 + \beta_1\, z_{(HV-LV)} + \beta_2\, z_{(HV+LV)} + \beta_3\, z_{(D-HV)} + \beta_4\, z_{(HV-LV)}\, z_{(D-HV)} + \varepsilon$

GLM2: Step 1, $\beta_0 + \beta_1\, z_{(HV-LV)} + \beta_2\, z_{(HV+LV)} + \varepsilon_1$

Step 2, $\beta_3 + \beta_4\, z_{(D-HV)} + \varepsilon_2$

GLM3: $\beta_0 + \beta_1\, z_{(HV-LV)} + \beta_2\, z_{(HV+LV)} + \beta_3\, z_{(D-HV)} + \varepsilon$

There are multiple ways to index the distractor value in the GLMs, such as D-HV, D-LV and D. Each of which should provide similar results because they are strongly correlated to each other ($r > 0.47$). However, the D-HV term was selected for easier comparison with the HV-LV effect and also with the distractor effects reported in previous studies (*Chau et al., 2014*; *Chau et al., 2020*). In each GLM, $z(x)$ refers to z-scoring of term $x$. GLM1 is identical to GLM1b of *Chau et al., 2020*. The variances of the HV and LV options are accounted by the $z_{(HV-LV)}$ and $z_{(HV+LV)}$ terms and the critical distractor effect was tested by the $z_{(D-HV)}$. In addition, the $z_{(HV-LV)}\, z_{(D-HV)}$ interaction term tested whether the distractor effect varied as a function of difficulty level (i.e. HV-LV). For cases with significant (HV-LV)(D-HV) effect, the data were median split by the HV-LV term and analysed using GLM2 to test the distractor effect on each half of the data. GLM2 involved a stepwise procedure to partial out the effects of HV and LV from the choice accuracy data. Then the distractor effects were tested on the residual $\varepsilon_1$ using the $z_{(D-HV)}$ term. GLM3 was applied to assay the impact of TMS on the distractor effect. It is a simplified version of GLM1 in which the $z_{(HV-LV)}\, z_{(D-HV)}$ interaction term was excluded. It was applied on the TMS and Non-TMS trials separately.

Each GLM was applied separately to each participant and each of the following conditions: TMS trials/Non-TMS trials in sessions in which the MIP/MT was targeted. Since the parietal neurons, including those from MIP, often have a response field that is selective to a small part of a contralateral space, the trials were further broken down into whether the distractor was displayed contralaterally/ipsilaterally to the stimulation, resulting in, at most, a total of eight conditions. The resulting beta values obtained from a GLM were then entered into a Site (MIP/MT) x Stimulation (TMS/Non-TMS) x Distractor Location (contralateral/ipsilateral) ANOVA, to test the impact of the different stimulation conditions on the relationship between stimulus values and accuracy. We hypothesised that MIP stimulation disrupts the negative distractor effect, while sparing the MIP-unrelated positive distractor effect, when the distractor was presented on the contralateral side. Therefore, MIP stimulation should result in a more positive relationship between distractor value and accuracy than control trials from the same session when the distractor was presented on the contralateral side. In addition, a more positive relationship between distractor value and accuracy should be apparent when contralateral distractor trials with MIP TMS are compared with any conditions in the control MT sessions.

## VBM Data Analysis

We set out to explore the relationship between local GM and the effect of TMS on decision-making. To this end, we used beta weights estimated by GLM3 used in the behavioural analysis, and subtracted the beta weights associated with the Non-TMS condition from the beta weights associated with the TMS condition (specifically: [MIP/TMS/Contralateral] – [MIP/Non-TMS/Contralateral]) for each predictor. The resulting differences were normalised and entered as explanatory variables in the VBM. This allowed us to test the relationship between local GM and the TMS effect on each predictor's impact on decision accuracy. We hypothesised that the difference in beta weights associated with the D-HV predictor (i.e. the impact of TMS on the distractor effect) is positively related to the size of parietal regions (i.e. the disruptive effect of TMS applied to MIP will be smaller in participants with larger MIPs because less of the volume of MIP will have been affected by the TMS). We repeated the same analysis in relation to the TMS effect associated with control MT stimulation (specifically: [MT/TMS/Contralateral] – [MT/Non-TMS/Contralateral]), but expected no effects.

## Eye-tracking Data Analysis

Apart from the MIP-related negative distractor effect, the MIP-unrelated positive distractor effect is associated with overt attention that is reflected by eye-movement (*Chau et al., 2014*; *Chau et al., 2020*). Here, we hypothesise that this effect is increased when TMS is applied over MIP, disrupting the positive distractor effect's antagonist. We therefore tested the impact of the presented stimuli on gaze shifts between the distractor stimulus (D) and the high-value (HV) option. We first tested all gaze shifts, that is all instances in which two immediately adjacent fixations included D and HV. To test directionality, we also repeated the analysis on gaze shifts from D to HV, and from HV to D separately.

The analysis of the eye-tracking data followed a similar process as the analysis of the behavioural data. GLM3 (predictors: HV-LV, HV +LV, D-HV; see also *Figure 3—figure supplement 3*), here using a normal distribution, was applied to predict gaze shifts. As in the behavioural analysis, the GLM was applied for each person and each condition separately, and a repeated-measures ANOVA (Site x Stimulation) was conducted on the resulting beta values (note that a Site x Simulation analysis was chosen for simplicity, but that a Site x Stimulation x Distractor Location analysis revealed qualitatively identical results).

Additionally, we explored the effect of gaze shifts between all possible options (HV to LV, HV to D, LV to HV, LV to D, D to LV, and D to HV) on accuracy using GLM4:

GLM4: $\beta_0 + \beta_1 \, z_{(HV\ to\ LV\ shift)} + \beta_2 \, z_{(HV\ to\ D\ shift)} + \beta_3 \, z_{(LV\ to\ HV\ shift)} + \beta_4 \, z_{(LV\ to\ D\ shift)} + \beta_5 \, z_{(D\ to\ LV\ shift)} + \beta_6 \, z_{(D\ to\ HV\ shift)} + \varepsilon.$

Since we did not have sufficient data to test these predictors on each condition, we tested their effect on accuracy across all Non-TMS trials. We were unable to identify sufficient gaze shifts in five participants and excluded these from the analysis. The resulting beta values of the remaining participants were then analysed using two-tailed one-sample t-tests.

## Acknowledgements

This work was supported by the Hong Kong Research Grants Council (15603517), the State Key Laboratory of Brain and Cognitive Sciences, The University of Hong Kong, and the Wellcome Trust grant (221794/Z/20/Z).

## Additional information

### Funding

| Funder | Grant reference number | Author |
|---|---|---|
| Hong Kong Research Grants Council | 15603517 | Bolton KH Chau |
| State Key Laboratory of Brain and Cognitive Science, The University of Hong Kong | | Bolton KH Chau |
| Wellcome Trust | 221794/Z/20/Z | Matthew FS Rushworth |

The funders had no role in study design, data collection and interpretation, or the decision to submit the work for publication. For the purpose of Open Access, the authors have applied a CC BY public copyright license to any Author Accepted Manuscript version arising from this submission.

### Author contributions

Carmen Kohl, Conceptualization, Data curation, Formal analysis, Supervision, Investigation, Methodology, Writing – original draft, Project administration, Writing – review and editing; Michelle XM Wong, Conceptualization, Data curation, Methodology, Project administration; Jing Jun Wong, Project administration, Writing – review and editing; Matthew FS Rushworth, Conceptualization, Formal analysis, Supervision, Methodology, Writing – original draft, Writing – review and editing; Bolton KH Chau, Conceptualization, Data curation, Formal analysis, Supervision, Funding acquisition, Investigation, Methodology, Writing – original draft, Project administration, Writing – review and editing

### Author ORCIDs
Jing Jun Wong http://orcid.org/0000-0001-5241-6997
Bolton KH Chau http://orcid.org/0000-0002-6854-5176

### Ethics
Human subjects: The experiment was approved by the Human Subjects Ethics Committee of the Hong Kong Polytechnic University and informed consent was obtained from all participants (HSEAR20151208001).

### Decision letter and Author response
Decision letter https://doi.org/10.7554/eLife.75007.sa1
Author response https://doi.org/10.7554/eLife.75007.sa2

## Additional files

### Supplementary files
• Transparent reporting form

### Data availability
The behavioural data, eye-tracking data, anonymized MRI grey matter volume data collected and analysed during the current study are available on Dryad: https://doi.org/10.5061/dryad.ngf1vhhzn.

The following dataset was generated:

| Author(s) | Year | Dataset title | Dataset URL | Database and Identifier |
|---|---|---|---|---|
| Kohl C, Wong MXM, Wong JJ, Rushworth MFS, Chau BKH | 2023 | Intraparietal stimulation disrupts negative distractor effects in human multi-alternative decision-making | https://doi.org/10.5061/dryad.ngf1vhhzn | Dryad Digital Repository, 10.5061/dryad.ngf1vhhzn |

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
