## [Editor Report]

This work presents fundamental findings elucidating the debate on how value-based choice behavior is influenced by seemingly irrelevant options (distractors). With convincing behavioral evidence following non-invasive brain stimulation, the authors provide support for the role of medial intraparietal cortex in divisive normalization in decision making. Given the importance of context effects as suboptimal violations of normative choice theories, this finding is significant and broadly relevant to psychologists, neuroscientists, and economists interested in decision making.

---

## [Decision Letter]

**Decision letter after peer review:**

Thank you for submitting your article "Intraparietal stimulation disrupts negative distractor effects in human multi-alternative decision-making" for consideration by *eLife*. Your article has been reviewed by 2 peer reviewers, and the evaluation has been overseen by a Reviewing Editor and Joshua Gold as the Senior Editor. The reviewers have opted to remain anonymous.

Essential revisions:

1) There was some concern about the lack of robustness of the TMS results and running an ANOVA on the betas from subject specific GLMs given that this ignores the estimated variance of the betas. In the follow up ANOVAs the standard errors from the individual GLMs are discarded, potentially impacting the results. In consultation the reviewers suggested potentially running a giant regression (all the data) with subject-specific effects for HV+LV and HV-LV and then a group level interaction on D-HV (including the site*location*TMS interactions). This would allow a simple t-test on the 3-way coefficient. This would essentially be a group-level GLM without subject-specific coefficients (this essentially averages over all subjects) and cluster standard errors at the subject level.

2) There is a strong assumption in the analyses that subjects make choices with perfect calculations of expected value (linear utility). However, it is well-known that people vary quite a bit from optimal EV calculations and have different risk preferences and biases that drive decisions. The authors need to show that their results hold with non-linear utility functions and/or individualized utility functions for each subject if possible.

3) There were several concerns about the interpretation of the results, justification of analysis choices, and potential limitations of the study given the design that need to be addressed (see reviews below).

*Reviewer #1 (Recommendations for the authors):*

This paper from Kohl and colleagues examines the biological basis of context-dependent decision-making, in which the preference for given options depends on factors beyond their intrinsic values. They authors target a specific form of context effect in trinary choice where unchosen options nevertheless shift the relative preference between the remaining two alternatives, a violation of traditional, normative theories of choice behavior. The existence and nature of these effects have been a recent topic of debate: some studies report negative effects (where distractors decrease choice accuracy in a value-dependent manner), some report positive effects (distractors increase accuracy), and some report no effect at all. Recent work from this same group showed that these disparate results can be reconciled if choosers exhibit both forms of context-dependence (in different regions of decision space defined by choice difficulty), proposing a composite model where prefrontal and parietal cortices mediated positive and negative effects, respectively.

Here, the authors present data replicating the coexistence of positive and negative distractor effects and, furthermore, show causal evidence for the role of intraparietal cortex – disrupting the medial intraparietal cortex (MIP) with transcranial magnetic stimulation selectively reduces negative distractor effects, with the degree of disruption correlating (negatively) with the size of MIP. The paper is clearly written and the results together support both the idea of a composite model, providing one of the only causal manipulations of such context effects. Overall the experimental approach is well reasoned and the results important, though there are some subtleties of analysis the authors should address, including the definition of regressors, the potential impact of nonlinear subjective values, and the interpretability/relevance of phantom decoy effects.

1) Definition and justification for regressors. The choice of regressor terms to test context-dependence could be better explained here, even if they follow past publications. Why is distracter value formalized as D-HV rather than D (e.g. in the interaction term (HV-LV)(D-HV))? Is there a particular reason to use D-HV instead of D, D-LV, or some other construct? And given this formalization, why do the authors use D in the interaction (HV+LV)D when testing the prediction of normalization (pg. 10).

2) Assumption of expected value choosers (linear utility). Given the crucial role of choice accuracy in the analyses, it is a bit concerning that correct choices are defined only by expected value calculations given the common finding of empirical risk preferences. The main question is whether any of the results would change if the choosers were in fact not expected value (risk neutral) choosers and choice performance was misspecified. The authors could reasonably argue that subjects are likely close to risk neutral given the small stakes, but given the central role of accuracy in the analyses, it is an important point to quantitatively establish. For example, the authors could either: (A) quantify a nonlinear utility function for individual subjects (either using effective binary trials where D=0, if they exist; or aggregating across all choices at all D values), or (B) show that the main results still hold assuming a reasonable range of nonlinear utility functions and redefined accuracy values.

3) Practical relevance. The task design using the ultimately unchoosable distractor is clever, allowing the examination of distractor effects in otherwise untestable scenarios (when D is high value and would be chosen over both HV and LV options). One issue is that the nature of the task – involving a mid-trial shift in choosable options – may elicit different internal dynamics and choices than a simple choice. In this regard, a more natural way to elicit distractor effects is a rank ordering approach (as in Dumbalska et al., 2020, PNAS); I certainly don't expect the authors to revise their task, but a discussion of the possible limitations would help. Aside from the unnatural dynamics, the use of the phantom decoy means that distractor effects are examined in many cases when they would not naturally occur (i.e. when the distractor would be selected). Do the main effects (dual distractor effects, MIP role in negative effects, morphometry results) still hold if the authors only include trials where D is lower than the LV and HV?

4) Support for Figure 3D (accuracy as function of D-HV in MIP TMS condition). The figure legend states that this panel supports an increase in accuracy with increasing distractor value, but the support for this statement isn't clear. From the graph, the trend in the MIP Contra condition is only mildly more evident that the other conditions, visually; more importantly, the main text does not present a quantitative analysis of a dependence on D-HV (e.g. regression) – it only presents the results of ANOVAs. Is there additional analysis that the authors meant to present?

5) Robustness of TMS effects. One statistical concern for the paper is that the primary TMS results are supported by effects with relatively marginal significance: three way interaction for D-HV p = 0.044, contralateral site x stimulation ANOVA p = 0.034, contralateral MIP stimulation ANOVA p = 0.049. This limitation is evident to the authors, as they avoid the additional splitting of data by difficulty and acknowledge the power issues. One finding in the authors' favor is that the effect is stronger with the inclusion of MIP size as a covariate. There is not much that we as reviewers can reasonably ask at this point, but perhaps a acknowledgement of this issue and a discussion of potential reasons and future plans to address it would be helpful.

*Reviewer #2 (Recommendations for the authors):*

This paper addresses a current debate on whether (and how) value-based choice behaviour is influenced by seemingly irrelevant options (distractors). Previous work has proposed an influence of distractor value on accuracy arising from a divisive normalization computation, which acts as a form of gain normalization on neural activity. There is evidence for such a computation in the lateral intra-parietal region of non-human primates.

The view being advanced by the paper is two-fold. First, that the effects of DN (a reduction in choice accuracy) should be most prominent when the magnitudes (sum) of options are low and decisions are difficult. And second, that this DN process takes place in parietal regions, and is competitive with some other process (perhaps in vmPFC) which induces positive distractor effect (increased accuracy).

The paper reports results from new experiments (of a previously studied-design) that assess the causal role of the medial intra-parietal area (MIP) on choice via TMS. The authors report that disruption of MIP increases choice accuracy. This effect is interpreted as disruption of the DN effect on choice. The experimental design is appropriate and the experiment and analysis are well-executed.

My main concerns with the paper are on the interpretation of the results and how they relate to the underlying theory being proposed. These concerns are essentially expositional. I would suggest a revision to address them.

1) The paper ascribes a negative distractor effect to DN, and a positive distractor effect to some other process (perhaps mutual inhibition). It is important to clarify what DN model the paper is considering, and what its predictions are. The theoretical statements on pg 6 lines 97-107 seem to be describing the model presented by Louie et al. in which the normalization term contains the non-weighted sum, but is not clear on the noise assumption. It is important to note a few issues here.

– The form of DN considered in Louie et al., 2013 is highly simplified. The perception literature suggests a number of more complicated functional forms with parameters (or perhaps even computations) that are tuned to the task. It would be VERY surprising if a choice process in cortex used a DN computation in which all weights and rectification parameters were set to 1. This is important, because statements about the role of the distractor in altering choice accuracy depend on this.

– DN can induce a positive distractor effect in some regions of decoy space. This is reported in the original Louie et al. paper, and is due to the Gaussian noise assumption (see Webb et al., 2020a, Figure 5 and Proposition 3).

– More broadly, where DN effects are strongest (as a function of option differences and magnitudes) also depends on the form of DN considered and the noise assumption. The paper is not very precise on the conditions required for the statements on pg 6 to be true. They can be verified analytically for a general form of DN and the Gumbel error. But a more general statement is difficult. Have the authors demonstrated this theoretical statement previously?

pg 6 lines 104-107. Whether positive distractor effects should dominate (depending on decision difficulty) also depends on a few factors. What is verifiable is that, under some assumptions, the influence of DN on choice ratios is smallest when the choice is easy (see above). Whether this leads to a positive distractor effect depends on the other unspecified process (like a MI model) and its relative influence. But this is not a prediction of DN, as implied in line 97, rather it is a prediction of the dual-route theory.

pg11 lines 201-202. It is not clear to me why the effect “reverses” since the main effect is not significant. Perhaps it is “pronounced” or “apparent” on difficult trials?

Figure 3D is a bit hard to interpret. Why is the data binned in to 4 bins? And not just a continuous regression reported? I suspect the figure is trying to demonstrate a positive trend as a function of D, but no statistical tests are run on these bins (and none seem significant). Perhaps the fitted values from the GLM could be generated as a function of D instead?

lines 394-395: What is the increase in gaze shifts between D and HV relative to? gaze shifts between D and LV? HV and LV?

pg 30 lines 540: It is not clear how these results are evidence for a causal relationship between mPFC and a positive distractor effect. Couldn’t the lesion of vmPFC just be creating substantial noise in the valuations, regardless of whether a negative distractor effect is operating. It seems odd to call this a “reduced positive distractor effect”. This matters for the discussion of the causal effect of vmPFC TMS below on pg 31. Assigning this effect to a relatively "stronger influence of DN" wouldn't be accurate, as it could simply reflect an increase in noise.

There is a limitation when using choice data alone to argue that TMS reduces normalization at the expense of some other process which has a positive distractor effect. Perhaps TMS alters the DN computation (i.e. its weights) in a way that induces a positive effect?

More broadly, the paper is not clear on how the dual-route theory operates. Is it an either/or process, so that when intra-parietal regions are TMS’d they are essentially “deactivated” and the decision is guided solely by vmPFC? If so, what process determines which “route” yields the decision in the control conditions? It would seem strange for this switching process to use value (i.e. difficulty) as the switching condition given that value is seemingly what is being constructed/determined. Or is the value process sequential, so that output from vmPFC feeds into parietal regions? If so, wouldn’t the TMS just be adding noise to the valuation process, reducing accuracy?

[Editors' note: further revisions were suggested prior to acceptance, as described below.]

Thank you for resubmitting your work entitled "Intraparietal stimulation disrupts negative distractor effects in human multi-alternative decision-making" for further consideration by *eLife*. Your revised article has been evaluated by Joshua Gold (Senior Editor) and a Reviewing Editor.

The manuscript has been improved but there are some remaining issues that need to be addressed, as outlined below:

1. Please address the outstanding concerns of Reviewer 2 (see below).

*Reviewer #2 (Recommendations for the authors):*

Overall the authors have responded sufficiently to previous comments, however, there is one issue that remains. I am not convinced by the new mixed-effects analysis in Figure 3-supplemental 1 that is intended to bolster the original ANOVA analysis of the TMS results because it is impossible to assess the new results.

The authors don't report the estimated coefficients in a table (only the coefficients and p-values for the variables of interest in Figure 3) and allude to a better fitting model when they include specific (un-reported) co-variates but don't appear to report this metric. This is non-standard when reporting structural model fits of choice behaviour and problematic because including RT in this regression where choice is the outcome variable is subject to a (likely high) endogeneity problem (it is correlated with the error term in the regression) therefore the other coefficients are likely biased (see discussion in Webb (2019) and Chiong et al., (forthcoming)). Without seeing the model results that don't include RT, it is impossible to assess which direction the bias goes. Please report all coefficients in these regressions, as well as the results for the model without RT as a regressor. If the authors want to include RT in the empirical analysis, they need to address the endogeneity problem by either estimating a dynamic model (like they propose with their race model, or finding a valid instrument using a control function approach).

Webb, R. The (Neural) Dynamics of Stochastic Choice. Management Science.i 65, 230-255 (2019).

Chiong, Shum, Webb, Chen. Combining Choice and Response Time Data: A Drift-Diffusion Model of Mobile Advertisements. Management Science. https://papers.ssrn.com/sol3/papers.cfm?abstract_id=3289386

---

## [Author Response]

Essential revisions:1) There was some concern about the lack of robustness of the TMS results and running an ANOVA on the betas from subject specific GLMs given that this ignores the estimated variance of the betas. In the follow up ANOVAs the standard errors from the individual GLMs are discarded, potentially impacting the results. In consultation the reviewers suggested potentially running a giant regression (all the data) with subject-specific effects for HV+LV and HV-LV and then a group level interaction on D-HV (including the site*location*TMS interactions). This would allow a simple t-test on the 3-way coefficient. This would essentially be a group-level GLM without subject-specific coefficients (this essentially averages over all subjects) and cluster standard errors at the subject level.

We thank the reviewers for the suggestions about improving the robustness of the TMS results. We noticed that the effects become most robust when we (1) follow the editor’s suggestion of collapsing all participants’ data using a logistic mixed-effects analysis, (2) replace the D-HV term by a D term, and (3) include reaction time as covariate. Specifically, the analysis included the terms of HV-LV, HV+LV, and RT that are grouped by subject along with the D term that is grouped by Site, Stimulation, and Location. Here, we observed a significant TMS effect (i.e. the distractor effect became more positive after TMS; *F*_1,16048_=13.87, *p*=0.000197) when the distractor was located on the contralateral side of space to the MIP stimulation. In addition, there was an absence of TMS effect when the distractor was located on the ipsilateral side of space to the MIP stimulation (*F*_1,16048_=0.0206, *p*=0.886) or when TMS was applied to the control MT region (contralateral: *F*_1,16048_=1.377, *p*=0.241; ipsilateral: *F*_1,16048_=1.468, *p*=0.226). We have now reported these results that show the TMS effect at a stronger significance level:

“Finally, the MIP-TMS effect was even more robust when we analysed the data from all participants together in a mix-effect model (Figure 3—figure supplement 1) …” (Lines 266-269)

2) There is a strong assumption in the analyses that subjects make choices with perfect calculations of expected value (linear utility). However, it is well-known that people vary quite a bit from optimal EV calculations and have different risk preferences and biases that drive decisions. The authors need to show that their results hold with non-linear utility functions and/or individualized utility functions for each subject if possible.

We acknowledge that non-linear utility functions are commonly used in this field to capture choices that are highly subjective. As suggested, we have now supplemented the analyses of decision making using a non-linear, individualized function to transform the “optimal” EVs. The key results remain significant. In particular, we first compared six utility functions and identified that the non-linear Prospect theory composite model provides the best account of participants’ choices. In particular, this model first applies the classical Prospect Theory to transform the reward magnitudes and probabilities in a non-linear manner (Kahneman and Tversky, 1979). These two attributes are then combined using both additive and multiplicative methods. Next, we applied the non-linearly-transformed values, repeated the analyses, and found that all the key results reported in Figure 3 remain significant (Figure 3—figure supplement 2).

In our revised manuscript we have continued to report the objective value results first in order to pre-empt any concerns that readers unfamiliar with subjective values might have. We have then followed up with a report of analyses based on subjective value estimates. The revised text now reads as follows:

“Finally, the MIP-TMS effect was even more robust when we analysed the data from all participants together in a mix-effect model (Figure 3—figure supplement 1) or when we considered that participants may evaluate choice attributes contributing to value in a non-linear manner (Figure 3—figure supplement 2).” (Lines 266-269)

3) There were several concerns about the interpretation of the results, justification of analysis choices, and potential limitations of the study given the design that need to be addressed (see reviews below).

Thank you for the collating the comments. We have now provided a point-by-point reply to each of the reviews below. In each case, we explain how we have revised the manuscript in light of the comments made.

Reviewer #1 (Recommendations for the authors):This paper from Kohl and colleagues examines the biological basis of context-dependent decision-making, in which the preference for given options depends on factors beyond their intrinsic values. They authors target a specific form of context effect in trinary choice where unchosen options nevertheless shift the relative preference between the remaining two alternatives, a violation of traditional, normative theories of choice behavior. The existence and nature of these effects have been a recent topic of debate: some studies report negative effects (where distractors decrease choice accuracy in a value-dependent manner), some report positive effects (distractors increase accuracy), and some report no effect at all. Recent work from this same group showed that these disparate results can be reconciled if choosers exhibit both forms of context-dependence (in different regions of decision space defined by choice difficulty), proposing a composite model where prefrontal and parietal cortices mediated positive and negative effects, respectively.Here, the authors present data replicating the coexistence of positive and negative distractor effects and, furthermore, show causal evidence for the role of intraparietal cortex – disrupting the medial intraparietal cortex (MIP) with transcranial magnetic stimulation selectively reduces negative distractor effects, with the degree of disruption correlating (negatively) with the size of MIP. The paper is clearly written and the results together support both the idea of a composite model, providing one of the only causal manipulations of such context effects. Overall the experimental approach is well reasoned and the results important, though there are some subtleties of analysis the authors should address, including the definition of regressors, the potential impact of nonlinear subjective values, and the interpretability/relevance of phantom decoy effects.

We are very pleased that the reviewer found the report to be potentially interesting.

1) Definition and justification for regressors. The choice of regressor terms to test context-dependence could be better explained here, even if they follow past publications. Why is distracter value formalized as D-HV rather than D (e.g. in the interaction term (HV-LV)(D-HV))? Is there a particular reason to use D-HV instead of D, D-LV, or some other construct? And given this formalization, why do the authors use D in the interaction (HV+LV)D when testing the prediction of normalization (pg. 10).

Thank you for the comment. In GLM1-3, we indexed the distractor value as D-HV and tested the effect of distractor in a hypothesis free approach – we allowed the models to freely fit a positive weight (that supports the mutual inhibition account) or a negative weight (that supports the divisive normalisation account) to the D-HV term. In the GLM included in Figure 2—figure supplement 1, we tested a number of distractor effects, including (HV+LV)D term that allowed us to test specific predictions made by the divisive normalisation account. Below, we explain why we employed the D-HV term and (HV+LV)D term in these analyses.

One important reason why the distractor value was indexed as D-HV is because it allows easy comparison with previous studies. In addition, it is important to consider other terms in the GLMs. Let’s consider GLM1 that includes the terms HV-LV, HV+LV, D-HV and (HV-LV)(D-HV) for predicting choice accuracy. The HV-LV term that describes how much the LV option is *relatively* poorer than the HV option; it should predict much of the variance in how like the participant is to choose HV rather than LV and, therefore, should be the most robust parameter. To describe distractor value in a similar format to the HV-LV term, we consider that either the *relative* term HV-D (as in Chau et al., *Nat Neurosci* 2014) or D-HV (as in Chau et al., *eLife* 2020) should be a better choice than the *absolute* term D. However, once that is done, it is not possible to include both D-HV and D-LV, in addition to HV-LV, in the same GLM because that will result in rank deficiency. In other words, if distractor value is included as a relative term it can only be included as one relative term, either D-HV or D-LV. We consider that D-HV is a better choice than D-LV because the HV term is supposedly more choice-relevant than the LV term. Despite these subtleties, the effects of the distractor should remain similar whether it is defined as D-HV, the mirror term HV-D, D-LV, or just D (see Chau et al., *eLife*, 2020 Figure 4 and Figure 4—figure supplement 1). These reasons for the use of the D-HV term in GLM1 should also be applicable to GLM2 and GLM3. We have now revised the Methods to explain this (Lines 717-725).

GLM1: β_0_+β_1_ z_(HV-LV)_+β_2_ z_(HV+LV)_+β_3_ z_(D-HV)_+β_4_ z_(HV-LV)_ z_(D-HV)_+ε

GLM2: Step1, β_0_+β_1_ z_(HV-LV)_+β_2_ z_(HV+LV)_+ε_1_

Step 2, β_3_+β_4_ z_(D-HV)_+ε_2_

GLM3: β_0_+β_1_ z_(HV-LV)_+β_2_ z_(HV+LV)_+ β_3_ z_(D-HV)_+ε

“There are multiple ways to index the distractor value in the GLMs, such as D-HV, D-LV and D. Each of which should provide similar results because they are strongly correlated with each other (*r* > 0.47). However, the D-HV term was selected for easier comparison with the HV-LV effect and also with the distractor effects reported in previous studies^6,22^.”

We describe an additional analysis that employed the (HV+LV)D term on Page 10 and Figure 2—figure supplement 1 to test further the predictions of the divisive normalisation account. Before that, we have shown in Figure 2d that a negative distractor effect predicted by divisive normalisation existed in our data. However, we notice that divisive normalisation should also predict that the distractor’s negative impact should be stronger if the total value of the HV and LV alternatives is smaller. This is because divisive normalisation depends on the total value HV+LV+D and the distractor’s absolute value D should have greater contribution to the degree of normalisation when HV+LV is smaller. Thus, the distractor value should be indexed as the absolute value D in this analysis. We have now revised the legend of Figure 2—figure supplement 1 to explain this:

“Figure 2—figure supplement 1. Participants showed smaller negative distractor effect when HV+LV was large. Divisive normalisation models predict that the size of the negative distractor effect is smaller when HV+LV is large, since the absolute distractor value D contributes less to the overall value of HV+LV+D. (a) To test whether this was the case in our data,…” (Lines 957960)

2) Assumption of expected value choosers (linear utility). Given the crucial role of choice accuracy in the analyses, it is a bit concerning that correct choices are defined only by expected value calculations given the common finding of empirical risk preferences. The main question is whether any of the results would change if the choosers were in fact not expected value (risk neutral) choosers and choice performance was misspecified. The authors could reasonably argue that subjects are likely close to risk neutral given the small stakes, but given the central role of accuracy in the analyses, it is an important point to quantitatively establish. For example, the authors could either: (A) quantify a nonlinear utility function for individual subjects (either using effective binary trials where D=0, if they exist; or aggregating across all choices at all D values), or (B) show that the main results still hold assuming a reasonable range of nonlinear utility functions and redefined accuracy values.

We acknowledge that non-linear utility functions are commonly used in this field to capture choices that are highly subjective. As suggested, we have now supplemented the analyses of decision making using a non-linear, individualized function to transform the “optimal” EVs. The key results remain significant. In particular, we first compared six utility functions and identified that the non-linear Prospect theory composite model provides the best account of participants’ choices. In particular, this model first applies the classical Prospect Theory to transform the reward magnitudes and probabilities in a non-linear manner (Kahneman and Tversky, 1979). These two attributes are then combined using both additive and multiplicative methods. Next, we applied the non-linearly-transformed values, repeated the analyses, and found that all the key results reported in Figure 3 remain significant (Figure 3—figure supplement 2).

In our revised manuscript we have continued to report the objective value results first in order to pre-empt any concerns that readers unfamiliar with subjective values might have. We have then followed up with a report of analyses based on subjective value estimates. The revised text now reads as follows:

“Finally, the MIP-TMS effect was even more robust when we analysed the data from all participants together in a mix-effect model (Figure 3—figure supplement 1) or when we considered that participants may evaluate choice attributes contributing to value in a non-linear manner (Figure 3—figure supplement 2).” (Lines 266-269)

3) Practical relevance. The task design using the ultimately unchoosable distractor is clever, allowing the examination of distractor effects in otherwise untestable scenarios (when D is high value and would be chosen over both HV and LV options). One issue is that the nature of the task – involving a mid-trial shift in choosable options – may elicit different internal dynamics and choices than a simple choice. In this regard, a more natural way to elicit distractor effects is a rank ordering approach (as in Dumbalska et al., 2020, PNAS); I certainly don't expect the authors to revise their task, but a discussion of the possible limitations would help. Aside from the unnatural dynamics, the use of the phantom decoy means that distractor effects are examined in many cases when they would not naturally occur (i.e. when the distractor would be selected). Do the main effects (dual distractor effects, MIP role in negative effects, morphometry results) still hold if the authors only include trials where D is lower than the LV and HV?

We agree with the reviewer that having an unchoosable distractor has the disadvantage of potentially inducing a “mid-trial shift”, even though it has the important advantage that it allowed us to examine the distractor effect in a parametric range that is beyond the HV and LV options. Note that, because of this concern, we revealed the identity of the distractor soon (in 0.1 second) after the options were presented in the current study and also in two preceding studies (Chau et al., *Nat Neurosci,* 2014 and *eLife*, 2020). In addition, in Chau et al., *Nat Neurosci,* 2014, we ran an extra experiment that involved three choosable options and defined the distractor as the worst option. In Supplementary Information 7, we reported that the results were similar – there was a negative (HV-LV)D interaction effect.

We followed the reviewer’s constructive suggestion of analyzing trials where the D was lowest in value compared to the HV and LV options. The results were broadly consistent with our original claims. However, before describing these results, we noticed that it is not feasible to run such an analysis using the key GLM1 (with HV-LV, HV+LV, D-HV, and (HV-LV)(D-HV) as regressors) and GLM3 (with HV-LV, HV+LV, and D-HV as regressors). By design, we optimized the study efficiency by decorrelating these key parameters. The collinearity between the regressors will become serious if we analyze only those trials where the distractor value was poorest (e.g. the correlation between HV-LV and (HV-LV)(D-HV) changed from *r*=-0.032 to *r*=-0.920; the correlation between HV-LV and D-HV changed from *r*=-0.240 to *r*=-0.431). We addressed this in the 2014 study mentioned in the previous paragraph by carefully redistributing the option values in an additional experiment so as to avoid collinearity when the distractor assumed lower values. Nevertheless, we notice that this can be resolved by running the GLM described in Figure 2—figure supplement 1, which involves the terms HV-LV, HV+LV, and D, as well as all twoway and three-way interactions. We performed the analysis and found that the distractor effects remained similar. These results are reported in the revised manuscript as follows:

“One may argue that the distractor was only irrelevant to choices when its value is smallest. An additional analysis that excluded trials where the D exceeded the value of LV or HV confirmed that the distractor effect remained comparable (Figure 2—figure supplement 1b).” (Lines 176-179).

4) Support for Figure 3D (accuracy as function of D-HV in MIP TMS condition). The figure legend states that this panel supports an increase in accuracy with increasing distractor value, but the support for this statement isn't clear. From the graph, the trend in the MIP Contra condition is only mildly more evident that the other conditions, visually; more importantly, the main text does not present a quantitative analysis of a dependence on D-HV (e.g. regression) – it only presents the results of ANOVAs. Is there additional analysis that the authors meant to present?

As suggested by another reviewer, we have now shown Figure 3D using the fitted results.

Hopefully the trend should look clearer now:

5) Robustness of TMS effects. One statistical concern for the paper is that the primary TMS results are supported by effects with relatively marginal significance: three way interaction for D-HV p = 0.044, contralateral site x stimulation ANOVA p = 0.034, contralateral MIP stimulation ANOVA p = 0.049. This limitation is evident to the authors, as they avoid the additional splitting of data by difficulty and acknowledge the power issues. One finding in the authors' favor is that the effect is stronger with the inclusion of MIP size as a covariate. There is not much that we as reviewers can reasonably ask at this point, but perhaps a acknowledgement of this issue and a discussion of potential reasons and future plans to address it would be helpful.

We thank the reviewer, as well as the editor for pointing out the issue of the significance level of the TMS effect. We noticed that the effects become most robust when we (1) follow the editor’s suggestion of collapsing all participants’ data using a logistic mixed-effects analysis, (2) replace the D-HV term by a D term, and (3) include reaction time as covariate. Specifically, the analysis included the terms of HV-LV, HV+LV, and RT that are grouped by subject along with the D term that is grouped by Site, Stimulation, and Location. Here, we observed a significant TMS effect (i.e. the distractor effect became more positive after TMS; *F*_1,16048_=13.87, *p*=0.000197) when the distractor was located on the contralateral side of space to the MIP stimulation. In addition, there was an absence of TMS effect when the distractor was located on the ipsilateral side of space to the MIP stimulation (*F*_1,16048_=0.0206, *p*=0.886) or when TMS was applied to the control MT region (contralateral: *F*_1,16048_=1.377, *p*=0.241; ipsilateral: *F*_1,16048_=1.468, *p*=0.226). We have now reported these results that show the TMS effect at a much better significance level:

“Finally, the MIP-TMS effect was even more robust when we analysed the data from all participants together in a mix-effect model (Figure 3—figure supplement 1) …” (Lines 266-269)

Reviewer #2 (Recommendations for the authors):This paper addresses a current debate on whether (and how) value-based choice behaviour is influenced by seemingly irrelevant options (distractors). Previous work has proposed an influence of distractor value on accuracy arising from a divisive normalization computation, which acts as a form of gain normalization on neural activity. There is evidence for such a computation in the lateral intra-parietal region of non-human primates.The view being advanced by the paper is two-fold. First, that the effects of DN (a reduction in choice accuracy) should be most prominent when the magnitudes (sum) of options are low and decisions are difficult. And second, that this DN process takes place in parietal regions, and is competitive with some other process (perhaps in vmPFC) which induces positive distractor effect (increased accuracy).The paper reports results from new experiments (of a previously studied-design) that assess the causal role of the medial intra-parietal area (MIP) on choice via TMS. The authors report that disruption of MIP increases choice accuracy. This effect is interpreted as disruption of the DN effect on choice. The experimental design is appropriate and the experiment and analysis are well-executed.My main concerns with the paper are on the interpretation of the results and how they relate to the underlying theory being proposed. These concerns are essentially expositional. I would suggest a revision to address them.

We are pleased that the reviewer found something of interest in our report. We understand the reviewer’s concerns and we have attempted to address them as we explain point-by-point below.

1) The paper ascribes a negative distractor effect to DN, and a positive distractor effect to some other process (perhaps mutual inhibition). It is important to clarify what DN model the paper is considering, and what its predictions are. The theoretical statements on pg 6 lines 97-107 seem to be describing the model presented by Louie et al. in which thenormalization term contains the non-weighted sum, but is not clear on the noise assumption. It is important to note a few issues here.

We agree with the reviewer that sometimes specifying the precise model is important to some debates. For example, Gluth and colleagues (*Nat Human Behav*, 2020) tried to argue that the divisive normalisation effect reported by Louie and colleagues (*PNAS*, 2013) was not replicable.

In a reply by Webb and colleagues (*Nat Human Behav,* 2020), it was demonstrated clearly that a more robust prediction could have been made had the divisive normalisation model included the right assumption about the noise distribution (i.e. a Gumbel distribution). We have attempted to make this clearer in the revised manuscript.

In one sense, the reviewer’s comments highlight the fact that DN models include two important elements. First, there is the divisive normalisation process itself. Despite the existence of variants of DN models, a general prediction is that neural responses and the resulting behavioural accuracy should decrease as the values of the alternatives are larger. A second important element, however, concerns the way in which noise is assumed to operate. An important consequence of different noise assumptions is, as the reviewer notes in their subsequent points below, on how the process of option comparison unfolds.

In this study, we do not intend to argue what should be the right specification of the divisive normalisation model, which is why we used more assumption-free terms (e.g. D-HV) to test for its impact. Our first main point is that divisive normalisation is clearly observable in our data and our second is that it can be disrupted by stimulating the parietal cortex. Nevertheless, we agree that we should remind readers that there are variants in how exactly the divisive normalisation model is specified. We have therefore made changes to the Introduction as shown below.

“One potential mechanism underlying this phenomenon is divisive normalisation. Based on the principles of normalisation observed in sensory systems^10,11^, Louie and colleagues^9^ proposed that the encoded value of a given stimulus corresponds to the stimulus’ absolute value divided by the weighted sum of the absolute values of co-occurring stimuli.. Despite differences in how exactly divisive normalisation is formalized (e.g. noise and weight assumptions), in general the hypothesis suggests that neural responses during decision-making or accuracy in decisionmaking are increased as the value of the best option increases and as the values of the remaining alternatives decrease. This hypothesis receives empirical support in a number of studies in humans and monkeys^8,12–16^. Recently, there has been support for this view from experiments showing that divisive normalisation can be applicable to multi-attribute choices, as normalisation can occur at the level of individual attributes before they are combined into an overall option value16,17.”

In addition, however, in the Discussion, we emphasize that the process of option comparison is commonly assumed to operate in the context of noise. Despite their differences, this is true in models that emphasize mutual inhibition between pools of neurons representing each choice (Chau et al., *Nature Neuroscience,* 2014), drift diffusion models of evidence accumulation (Chau et al., *eLife*, 2020), and in DN models (Webb, *Management Science*, 2020). In all cases, it is consideration of the operation of this comparison process in the context of noise that makes it possible to account for positive distractor effects. We have made changes to the Discussion along the lines shown below (Lines 501-521).

“It is important to note that a positive distractor effect can be induced in some regions of distractor space in divisive normalisation-based models^8,16^. This occurs because, as a noisy representation of the distractor increases in value, it is disproportionately more likely to be chosen than the LV option as opposed to the HV option. Whether this effect exerts a large influence in the current study is unclear given that participants are instructed never to choose the distractor. It is, nevertheless, the case that the distractor is, albeit infrequently, chosen by participants. Despite their differences, it is notable across a range of models that it is the operation of a choice comparison process in the context of noise that makes it possible to account for positive distractor effects. This is true whether it is a recurrent neural network model emphasizing mutual inhibition between pools of neurons representing each choice^6^, diffusion models of evidence accumulation for choice selection^22^, or even in certain parts of decision space in divisive normalisation^16^. While some models place greater emphasis on the presence of noise in the comparison process, other emphasize noise in the representation of the choices themselves. It is possible to envisage models in which the asymmetric impact of a noisy distractor representation on rejection of the LV option is greatest when HV and LV options are far apart in value. The empirical findings reported here and elsewhere^6,22^, however, suggest that the greatest impact occurs when HV and LV are close in value and it is difficult to choose between them. The high value distractor augments and protracts the comparison process during these difficult choices so that HV is more likely to be chosen.”

– The form of DN considered in Louie et al., 2013 is highly simplified. The perception literature suggests a number of more complicated functional forms with parameters (or perhaps even computations) that are tuned to the task. It would be VERY surprising if a choice process in cortex used a DN computation in which all weights and rectification parameters were set to 1. This is important, because statements about the role of the distractor in altering choice accuracy depend on this.– DN can induce a positive distractor effect in some regions of decoy space. This is reported in the original Louie et al. paper, and is due to the Gaussian noise assumption (see Webb et al., 2020a, Figure 5 and Proposition 3).

As suggested, we have noted positive distractor effects can be present even in DN models. in the Discussion section in the revised manuscript as follows (Lines 501-521):

“It is important to note that a positive distractor effect can be induced in some regions of distractor space in divisive normalisation-based models^8,16^. This occurs because, as a noisy representation of the distractor increases in value, it is disproportionately more likely to be chosen than the LV option as opposed to the HV option. Whether this effect exerts a large influence in the current study is unclear given that participants are instructed never to choose the distractor. It is, nevertheless, the case that the distractor is, albeit infrequently, chosen by participants. Despite their differences, it is notable across a range of models that it is the operation of a choice comparison process in the context of noise that makes it possible to account for positive distractor effects. This is true whether it is a recurrent neural network model emphasizing mutual inhibition between pools of neurons representing each choice^6^, diffusion models of evidence accumulation for choice selection^22^, or even in certain parts of decision space in divisive normalisation^16^. While some models place greater emphasis on the presence of noise in the comparison process, other emphasize noise in the representation of the choices themselves. It is possible to envisage models in which the asymmetric impact of a noisy distractor representation on rejection of the LV option is greatest when HV and LV options are far apart in value. The empirical findings reported here and elsewhere^6,22^, however, suggest that the greatest impact occurs when HV and LV are close in value and it is difficult to choose between them. The high value distractor augments and protracts the comparison process during these difficult choices so that HV is more likely to be chosen.”

– More broadly, where DN effects are strongest (as a function of option differences and magnitudes) also depends on the form of DN considered and the noise assumption. The paper is not very precise on the conditions required for the statements on pg 6 to be true. They can be verified analytically for a general form of DN and the Gumbel error. But a more general statement is difficult. Have the authors demonstrated this theoretical statement previously?

We understand the point being made by the reviewer. As noted already we have attempted to address some of these concerns by making changes to the Discussion (Lines 501-521).

“It is important to note that a positive distractor effect can be induced in some regions of distractor space in divisive normalisation-based models^8,16^. This occurs because, as a noisy representation of the distractor increases in value, it is disproportionately more likely to be chosen than the LV option as opposed to the HV option. Whether this effect exerts a large influence in the current study is unclear given that participants are instructed never to choose the distractor. It is, nevertheless, the case that the distractor is, albeit infrequently, chosen by participants. Despite their differences, it is notable across a range of models that it is the operation of a choice comparison process in the context of noise that makes it possible to account for positive distractor effects. This is true whether it is a recurrent neural network model emphasizing mutual inhibition between pools of neurons representing each choice^6^, diffusion models of evidence accumulation for choice selection^22^, or even in certain parts of decision space in divisive normalisation^16^. While some models place greater emphasis on the presence of noise in the comparison process, other emphasize noise in the representation of the choices themselves. It is possible to envisage models in which the asymmetric impact of a noisy distractor representation on rejection of the LV option is greatest when HV and LV options are far apart in value. The empirical findings reported here and elsewhere^6,22^, however, suggest that the greatest impact occurs when HV and LV are close in value and it is difficult to choose between them. The high value distractor augments and protracts the comparison process during these difficult choices so that HV is more likely to be chosen.”

pg 6 lines 104-107. Whether positive distractor effects should dominate (depending on decision difficulty) also depends on a few factors. What is verifiable is that, under some assumptions, the influence of DN on choice ratios is smallest when the choice is easy (see above). Whether this leads to a positive distractor effect depends on the other unspecified process (like a MI model) and its relative influence. But this is not a prediction of DN, as implied in line 97, rather it is a prediction of the dual-route theory.

We agree with the reviewer that some of the predictions previously described in the aforementioned paragraph are pertinent to the dual-route model, instead of divisive normalisation *per se*. We have now revised the corresponding section (Lines 93-113) to make this point clearer.

“Recently, however, it has been argued that neither the negative distractor effects predicted by divisive normalisation models nor the positive distractor effects predicted by mutual inhibition models are robust ^19,20^. One way of reconciling these disparate points of view (positive distractor effects; negative distractor effects; no distractor effects), however, is the notion that, in fact, both positive and negative distractor effects occur but predominate to different degrees in different circumstances. This can be achieved by having a dual-route model that contains both a “divisive normalisation” component and a “mutual inhibition” component that run in parallel for making a decision in a race^6^.

For example, careful consideration of the dual-route model suggests that the negative influence of the distractor should be most prominent in certain parts of a “decision space” defined by two dimensions –the total sum and the difference in values of the options ^21^. Firstly, in the divisive normalisation component, the negative impact caused by variance in distractor value should be greatest when the total sum of option values is low. In accordance with this prediction, distractors reliably exert a significant and negative effect on decision-making, when the sum of the values of the choosable options is small ^21^. Secondly, positive distractor effects should predominate in the parts of the decision space in which the values of both choosable options are close and decisions are difficult but the opposite should happen when decisions are easy to make because the choice option values are far apart. Both positive and negative distractor effects are apparent even in data sets in which they have been claimed to be absent ^6,21^.”

pg11 lines 201-202. It is not clear to me why the effect “reverses” since the main effect is not significant. Perhaps it is “pronounced” or “apparent” on difficult trials?

This comment is related to Figure 2. The combination of the absence of a D-HV main effect and the presence of a (HV-LV)(D-HV) interaction effect can be interpreted as indicating that the DHV effect is reversed as a function of the HV-LV term. In other words, the D-HV effect changes from positive to negative (or in the opposite direction) in different portions of the data such that they cancel out each other in the main effect. This is the case when we estimated the D-HV effects on hard (small HV-LV) trials and easy (large HV-LV) trials separately in Figure 2d. We have now revised these lines, such that it may be easier to follow:

“(c) GLM1 revealed that there was a negative (HV-LV)(D-HV) effect on accuracy, suggesting that the distractor effect (i.e. D-HV) varied as a function of difficulty (i.e. HV-LV). (d) A follow-up analysis on the (HV-LV)(D-HV) interaction using GLM2 showed that the distractor effect on accuracy was positive on hard trials and it was negative on easy trials.” (Lines 950-955)

Figure 3D is a bit hard to interpret. Why is the data binned in to 4 bins? And not just a continuous regression reported? I suspect the figure is trying to demonstrate a positive trend as a function of D, but no statistical tests are run on these bins (and none seem significant). Perhaps the fitted values from the GLM could be generated as a function of D instead?

We would like to thank the reviewer for the suggestion. We have now replaced Figure 3D by using the fitted values:

lines 394-395: What is the increase in gaze shifts between D and HV relative to? gaze shifts between D and LV? HV and LV?

These lines describe our previous findings that there is a positive correlation between distractor value and gaze shift between D and HV. In other words, it is a comparison between trials with smaller and larger distractor values, rather than a comparison between trials involving D and HV gaze shifts versus other types of gaze shifts. We have revised these lines and hopefully the clarity is now improved:

“Chau et al. (2020) showed that there is a positive correlation between distractor value and gaze shift between the D and HV options. As such, larger distractor values are associated with more D-and-HV gaze shifts and, ultimately, more accurate HV choices are made.” (Lines 347-350).

pg 30 lines 540: It is not clear how these results are evidence for a causal relationship between mPFC and a positive distractor effect. Couldn’t the lesion of vmPFC just be creating substantial noise in the valuations, regardless of whether a negative distractor effect is operating. It seems odd to call this a “reduced positive distractor effect”. This matters for the discussion of the causal effect of vmPFC TMS below on pg 31. Assigning this effect to a relatively "stronger influence of DN" wouldn't be accurate, as it could simply reflect an increase in noise.

Thank you for the comment. Although we would first like to first point out that this comment most directly relates to the discussion of a hypothetical case of perturbing the vmPFC and that it is arguably less directly related to the current empirical findings that concerns the parietal cortex, we understand that this issue is of some general interest. We have therefore tried to make changes to the manuscript. Arguably, the issues related to this comment are three-fold in nature. First, what is the consequence for choice behaviour of a vmPFC lesion? Second, what are the predicted effects of TMS were it possible to apply it to disrupt the vmPFC? Third, can the distractor effect observed in individuals with vmPFC lesion be explained by any changes in noise level?

First, to explain the effects associated with vmPFC lesion, on Page 30, we described evidence from two studies by Noonan and colleagues that involved testing the distractor effect in humans or monkeys. Both studies showed that a negative distractor effect was only present in individuals with vmPFC lesions. There was an absence of an overall distractor effect in control individuals, including those with a lesion in other areas of the prefrontal cortex. The findings of the current manuscript and Chau et al. (2020) demonstrate that positive and negative distractor effects co-exist in different parts of the decision space suggesting that the two effects can, in some circumstances, cancel out each other if the decision space is not considered in the analysis. We have revised Lines 477 to 484 to make this clearer.

“In contrast, a causal relation between the vmPFC and a positive distractor effect has been observed in both humans and monkeys ^28,29^. In both species, individuals with vmPFC lesions, unlike control or pre-operative individuals, were less accurate in their decisions when high-value distractors were presented. In vmPFC lesion individuals, the manifestation of a negative distractor effect is possibly due to the reduction of the opposing positive distractor effect (Figure 6c). Collectively, these findings demonstrate that positive and negative distractor effects are causally related to the neural processes in vmPFC and parietal cortex respectively.”

Second, in the Discussion section, we predict the behavioural effect of vmPFC-TMS based on the lesion studies by Noonan and colleagues, although it is technically difficult to apply TMS to the vmPFC because of its distance from the scalp. Often, TMS is considered as a virtual lesion as it disrupts brain activity. Hence, we expect that vmPFC-TMS will result in behavioural effects that are similar to those observed in vmPFC lesion patients. We do not expect that vmPFC-TMS would cause any changes in the negative distractor effect nor to any changes in the parietal cortex, but because the counteracting positive distractor effect is disrupted, then the overall behavioural impact should be that the distractor effect should become more negative (i.e. less positive). Nevertheless, it is possible that this argument was not explained clearly in the first draft of our manuscript. We have now revised Lines 529-542:

“There are two limitations in the current study. First, we did not directly test whether the positive distractor effect is generated in the vmPFC. Second, an alternative explanation for the presence of a positive distractor effect after MIP-TMS is that it is due to a direct change in the divisive normalisation computation in the MIP itself. We argue, however, that another interpretation is that by perturbing the MIP and reducing its associated negative distractor effect then this allows for stronger expression of the positive distractor effect associated with the vmPFC. This link is based on findings of previous studies that suggest a relationship between the positive distractor effect and vmPFC ^6^ and the greater prominence of negative distractor effects after vmPFC is damaged ^28,29^. A future study which includes stimulation of the vmPFC could examine this interpretation further. Note however, that TMS over frontal regions can cause discomfort in participants and would require an adjustment of the stimulation protocol. Perhaps most critically, it may not even be possible for TMS effects to be induced at the depth below the scalp at which vmPFC lies.”

Third, the reviewer questioned whether the negative distractor effect observed in individuals with vmPFC lesions could be explained simply by a greater noise level. To the best of our knowledge, it is not possible to explain the findings reported by Noonan and colleagues just by a noise model without considering additional mechanisms such as mutual inhibition. In a noise model (Gaussian or Gumbel), as the reviewer pointed out in Comment 1, the distractor “steals” density from the second best option and produces a “positive distractor effect”. If it is assumed that vmPFC lesion simply amplifies the noise, then potentially an even greater positive distractor effect (and smaller negative distractor effect) might be expected in the choices of humans or monkeys with vmPFC lesions. However, the empirical findings reported by Noonan and colleagues are the other way round; after vmPFC lesions, negative distractor effects are especially prominent.

There is a limitation when using choice data alone to argue that TMS reduces normalization at the expense of some other process which has a positive distractor effect. Perhaps TMS alters the DN computation (i.e. its weights) in a way that induces a positive effect?More broadly, the paper is not clear on how the dual-route theory operates. Is it an either/or process, so that when intra-parietal regions are TMS’d they are essentially “deactivated” and the decision is guided solely by vmPFC? If so, what process determines which “route” yields the decision in the control conditions? It would seem strange for this switching process to use value (i.e. difficulty) as the switching condition given that value is seemingly what is being constructed/determined. Or is the value process sequential, so that output from vmPFC feeds into parietal regions? If so, wouldn’t the TMS just be adding noise to the valuation process, reducing accuracy?

The current manuscript argues parietal cortex generates the negative distractor effect. To identify the effect clearly, however, it is essential to notice that both positive and negative distractor effects co-exist, for example, as in the form suggested by the dual-route model. Both effects potentially counteract each other so that additional care is required to isolate the negative distractor effect and then to test its link with the parietal cortex. We are reluctant to claim that, after parietal TMS, a decision is guided only by the vmPFC. However, at least the empirical findings are consistent with the idea that two distractor effects are present in decision making and applying TMS to the parietal cortex disrupts one effect and spares the other.

More generally, we note that the reviewer’s point about how the brain “manages” the presence of multiple selection processes in multiple brain regions is an important one, but it is an important problem for the whole field in which it is clear that there are multiple selection processes spanning parietal, premotor, cingulate, and ventromedial prefrontal cortical areas, as well as subcortical areas. Although it is a question that goes beyond what can be addressed with the current data set, we are generally attracted by the idea that multiple selection processes co-occur at the same time in the absence of an overall top-down controller. Instead, we imagine that what determines the likelihood that a particular selection mechanism will guide a given choice is the speed with which it operates in a given situation (Chau et al., 2014) and the certainty of the evaluations that it makes (Lines 434-443).

“These variations in the distractor effect are captured by a dual-route model that incorporates both a divisive normalisation mechanism and a mutual inhibition mechanism in separate decision routes that compete in a parallel race^22^. It is clear that multiple brain mechanisms are concerned with choice selection and that these span not just parietal cortex and vmPFC but also include orbitofrontal cortex, premotor cortex, and cingulate cortex as well as subcortical regions^28–31^. What determines which mechanism predominates in a given setting is likely to be related to the speed with which it operates in a given situation and with a given input^6^ and the certainty of the evaluations that it makes. The current results again suggest an association between these different routes and different brain regions.”

[Editors' note: further revisions were suggested prior to acceptance, as described below.]

The manuscript has been improved but there are some remaining issues that need to be addressed, as outlined below:1. Please address the outstanding concerns of Reviewer 2 (see below).Reviewer #2 (Recommendations for the authors):Overall the authors have responded sufficiently to previous comments, however, there is one issue that remains. I am not convinced by the new mixed-effects analysis in Figure 3-supplemental 1 that is intended to bolster the original ANOVA analysis of the TMS results because it is impossible to assess the new results.The authors don't report the estimated coefficients in a table (only the coefficients and p-values for the variables of interest in Figure 3) and allude to a better fitting model when they include specific (un-reported) co-variates but don't appear to report this metric. This is non-standard when reporting structural model fits of choice behaviour and problematic because including RT in this regression where choice is the outcome variable is subject to a (likely high) endogeneity problem (it is correlated with the error term in the regression) therefore the other coefficients are likely biased (see discussion in Webb (2019) and Chiong et al., (forthcoming)). Without seeing the model results that don't include RT, it is impossible to assess which direction the bias goes. Please report all coefficients in these regressions, as well as the results for the model without RT as a regressor. If the authors want to include RT in the empirical analysis, they need to address the endogeneity problem by either estimating a dynamic model (like they propose with their race model, or finding a valid instrument using a control function approach).Webb, R. The (Neural) Dynamics of Stochastic Choice. Management Science.i 65, 230-255 (2019).Chiong, Shum, Webb, Chen. Combining Choice and Response Time Data: A Drift-Diffusion Model of Mobile Advertisements. Management Science. https://papers.ssrn.com/sol3/papers.cfm?abstract_id=3289386

We are glad that the reviewer considers that the comments were sufficiently well responded to, except one that is related to Figure 3-supplement 1. We agree that for transparent reporting, we should have shown all coefficients of the model. Those coefficients that were previously not shown were related to the HV-LV and HV+LV of all the 31 participants. We realize that it would be a very long list of coefficients if they are tabulated and therefore, we think that it would be more intuitive to show them in a figure (now shown in panel b). In addition, we also agree including an RT term in the model could be controversial. Perhaps the most straightforward approach is to include both versions of the mixed-effects model – both with and without the RT term. Now we show the results of the model without the RT term, as suggested by the reviewer, in Figure 3—figure supplement 1a and b. We show those of the model with the RT term in Figure 3—figure supplement 1c and d. Critically, the results of both models were similar – showing that the TMS effect was robust when the distractor was located on the contralateral side of space to the MIP stimulation.

Once again, we would like to thank the reviewer for the constructive comments.